# An oscillating computational model can track pseudo-rhythmic speech by using linguistic predictions

**Sanne ten Oever[1,2,3]\*, Andrea E Martin[1,2]**

[1]Language and Computation in Neural Systems group, Max Planck Institute for Psycholinguistics, Nijmegen, Netherlands; [2]Donders Centre for Cognitive Neuroimaging, Radboud University, Nijmegen, Netherlands; [3]Department of Cognitive Neuroscience, Faculty of Psychology and Neuroscience, Maastricht University, Maastricht, Netherlands

**Abstract** Neuronal oscillations putatively track speech in order to optimize sensory processing. However, it is unclear how isochronous brain oscillations can track pseudo-rhythmic speech input. Here we propose that oscillations can track pseudo-rhythmic speech when considering that speech time is dependent on content-based predictions flowing from internal language models. We show that temporal dynamics of speech are dependent on the predictability of words in a sentence. A computational model including oscillations, feedback, and inhibition is able to track pseudo-rhythmic speech input. As the model processes, it generates temporal phase codes, which are a candidate mechanism for carrying information forward in time. The model is optimally sensitive to the natural temporal speech dynamics and can explain empirical data on temporal speech illusions. Our results suggest that speech tracking does not have to rely only on the acoustics but could also exploit ongoing interactions between oscillations and constraints flowing from internal language models.

**\*For correspondence:**
sanne.tenoever@mpi.nl

**Competing interests:** The authors declare that no competing interests exist.

## Introduction

Speech is a biological signal that is characterized by a plethora of temporal information. The temporal relationship between subsequent speech units allows for the online tracking of speech in order to optimize processing at relevant moments in time (*Jones and Boltz, 1989*; *Large and Jones, 1999*; *Giraud and Poeppel, 2012*; *Ghitza and Greenberg, 2009*; *Ding et al., 2017*; *Arvaniti, 2009*; *Poeppel, 2003*). Neural oscillations are a putative index of such tracking (*Giraud and Poeppel, 2012*; *Schroeder and Lakatos, 2009*). The existing evidence for neural tracking of the speech envelope is consistent with such a functional interpretation (*Luo et al., 2013*; *Keitel et al., 2018*). In these accounts, the most excitable optimal phase of an oscillation is aligned with the most informative time point within a rhythmic input stream (*Schroeder and Lakatos, 2009*; *Lakatos et al., 2008*; *Henry and Obleser, 2012*; *Herrmann et al., 2013*; *Obleser and Kayser, 2019*). However, the range of onset time difference between speech units seems more variable than fixed oscillations can account for (*Rimmele et al., 2018*; *Nolan and Jeon, 2014*; *Jadoul et al., 2016*). As such, it remains an open question how is it possible that oscillations can track a signal that is at best only pseudo-rhythmic (*Nolan and Jeon, 2014*).

Oscillatory accounts tend to focus on the prediction in the sense of predicting 'when', rather than predicting 'what': oscillations function to align the optimal moment of processing given that timing is predictable in a rhythmic input structure. If rhythmicity in the input stream is violated, oscillations must be modulated to retain optimal alignment to incoming information. This can be achieved through phase resets (*Rimmele et al., 2018*; *Meyer, 2018*), direct coupling of the acoustics to

oscillations (*Poeppel and Assaneo, 2020*), or the use of many oscillators at different frequencies (*Large and Jones, 1999*). However, the optimal or effective time of processing stimulus input might not only depend on when you predict something to occur, but also depend on what stimulus is actually being processed (*Ten Oever et al., 2013*; *Martin, 2016*; *Rosen, 1992*; *deen et al., 2017*).

What and when are not independent, and certainly not from the brain's-eye-view. If continuous input arrives to a node in an oscillatory network, the exact phase at which this node reaches threshold activation does not only depend on the strength of the input, but also depend on how sensitive this node was to start with. Sensitivity of a node in a language network (or any neural network) is naturally affected by predictions in the what domain generated by an internal language model (*Martin, 2020*; *Marslen-Wilson, 1987*; *Lau et al., 2008*; *Nieuwland, 2019*). We define internal language model as the individually acquired statistical and structural knowledge of language stored in the brain. A virtue of such an internal language model is that it can predict the most likely future input based on the currently presented speech information. If a language model creates strong predictions, we call it a strong model. In contrast, a weak model creates no or little predictions about future input (note that the strength of individual predictions depends not only on the capability of the system to create a prediction, but also on the available information). If a node represents a speech unit that is likely to be spoken next, a strong internal language model will sensitize this node and it will therefore be active earlier, that is, on a less excitable phase of the oscillation. In the domain of working memory, this type of phase precession has been shown in rat hippocampus (*O'Keefe and Recce, 1993*; *Malhotra et al., 2012*) and more recently in human electroencephalography (*Bahramisharif et al., 2018*). In speech, phase of activation and perceived content are also associated (*Ten Oever and Sack, 2015*; *Kayser et al., 2016*; *Di Liberto et al., 2015*; *Ten Oever et al., 2016*; *Thézé et al., 2020*), and phase has been implicated in tracking of higher-level linguistic structure (*Meyer, 2018*; *Brennan and Martin, 2020*; *Kaufeld et al., 2020a*). However, the direct link between phase and the predictability flowing from a language model has yet to be established.

The time of speaking/speed of processing is not only a consequence of how predictable a speech unit is within a stream, but also a cue for the interpretation of this unit. For example, phoneme categorization depends on timing (e.g., voice onsets, difference between voiced and unvoiced phonemes), and there are timing constraints on syllable durations (e.g., the theta syllable *Poeppel and Assaneo, 2020*; *Ghitza, 2013*) that affect intelligibility (*Ghitza, 2012*). Even the delay between mouth movements and speech audio can influence syllabic categorizations (*Ten Oever et al., 2013*). Most oscillatory models use oscillations for parsing, but not as a temporal code for content (*Panzeri et al., 2015*; *Kayser et al., 2009*; *Mehta et al., 2002*; *Lisman and Jensen, 2013*). However, the time or phase of presentation does influence content perception. This is evident from two temporal speech phenomena. In the first phenomena, the interpretation of an ambiguous short /ɑ/ or long vowel /a:/ depends on speech rate (in Dutch; *Reinisch and Sjerps, 2013*; *Kösem et al., 2018*; *Bosker and Reinisch, 2015*). Specifically, when speech rates are fast the stimulus is interpreted as a long vowel and vice versa for slow rates. However, modulating the entrainment rate effectively changes the phase at which the target stimulus – which is presented at a constant speech rate – arrives (but this could not be confirmed in *Bosker and Kösem, 2017*). A second speech phenomena shows the direct phase-dependency of content (*Ten Oever and Sack, 2015*; *Ten Oever et al., 2016*). Ambiguous /da/-/ga/ stimuli will be interpreted as a /da/ on one phase and a /ga/ on another phase. This was confirmed in both a EEG and a behavioral study. An oscillatory theory of speech tracking should account for how temporal properties in the input stream can alter what is perceived.

In the speech production literature, there is strong evidence that the onset times (as well as duration) of an uttered word is modulated by the frequency of that word in the language (*O'Malley and Besner, 2008*; *Monsell, 1991*; *Monsell et al., 1989*; *Powers, 1998*; *Piantadosi, 2014*) showing that internal language models modulate the access to or sensitivity of a word node (*Martin, 2020*; *Hagoort, 2017*). This word-frequency effect relates to the access to a single word. However, it is likely that during ongoing speech internal language models use the full context to estimate upcoming words (*Beattie and Butterworth, 1979*; *Pluymaekers et al., 2005a*; *Lehiste, 1972*). If so, the predictability of a word in context should provide additional modulations on speech time. Therefore, we predict that words with a high predictability in the producer's language model should be uttered relatively early. In this way, word-to-word onset times map to the predictability level of that word within the internal model. Thus, not only the processing time depends on the predictability of a

word (faster processing for predictable words; see *Gwilliams et al., 2020*; *Deacon et al., 1995*, and *Aubanel and Schwartz, 2020* showing that speech time in noise matters), but also the production time (earlier uttering of predicted words).

Language comprehension involves the mapping of speech units from a producer's internal model to the speech units of the receiver's internal model. In other words, one will only understand what someone else is writing or saying if one's language model is sufficiently similar to the speakers (and if we speak in Dutch, fewer people will understand us). If the producer's and receiver's internal language model have roughly matching top-down constrains, they should similarly influence the speed of processing (either in production or perception; *Figure 1A–C*). Therefore, if predictable words arrive earlier (due to high predictability in the producer's internal model), the receiver also expects the content of this word to match one of the more predictable ones from their own internal model (*Figure 1C*). Thus, the phase of arrival depends on the internal model of the producer and the expected phase of arrival depends on the internal model of the receiver (*Figure 1D*). If this is true, pseudo-rhythmicity is fully natural to the brain, and it provides a means to use time or arrival phase as a content indicator. It also allows the receiver to be sensitive to less predictable words when they arrive relatively late. Current oscillatory models of speech parsing do not integrate the constraints flowing from an internal linguistic model into the temporal structure of the brain response. It is therefore an open question whether the oscillatory model the brain employs is actually attuned to the temporal variations in natural speech.

Here, we propose that neural oscillations can track pseudo-rhythmic speech by taking into account that speech timing is a function of linguistic constrains. As such we need to demonstrate that speech statistics are influenced by linguistic constrains as well as showing how oscillations can be sensitive to this property in speech. We approach this hypothesis as follows: First, we demonstrate that in natural speech timing depends on linguistics predictions (*temporal speech properties*). Then, we model how oscillations can be sensitive to these linguistic predictions (*modeling speech tracking*). Finally, we validate that this model is optimally sensitive to the natural temporal properties in speech and displays temporal speech illusions (*model validation*). Our results reveal that tracking of speech needs to be viewed as an interaction between ongoing oscillations as well as constraints flowing from an internal language model (*Martin, 2016*; *Martin, 2020*). In this way, oscillations do not have to shift their phase after every speech unit and can remain at a relatively stable frequency as long as the internal model of the speaker matches the internal model of the perceiver.

## Results

### Temporal speech properties
#### Word frequency influences word duration
To extract the temporal properties in naturally spoken speech we used the Corpus Gesproken Nederlands (CGN; [Version 2.0.3; 2014]). This corpus consists of elaborated annotations of over 900 hr of spoken Dutch and Flemish words. We focus here on the subset of the data of which onset and offset timings were manually annotated at the word level in Dutch. Cleaning of the data included removing all dashes and backslashes. Only words were included that were part of a Dutch word2vec embedding (github.com/coosto/dutch-word-embeddings; *Nieuwenhuijse, 2018*; needed for later modeling) and required to have a frequency of at least 10 in the corpus. All other words were replaced with an <unknown> label. This resulted in 574,726 annotated words with 3096 unique words. Two thousand and forty-eight of the words were recognized in the Dutch Wordforms database in CELEX (Version 3.1) in order to extract the word frequency as well as the number of syllables per word (later needed to fit a regression model). Mean word duration was 0.392 s, with an average standard deviation of 0.094 s (*Figure 2—figure supplement 1*). By splitting up the data in sequences of 10 sequential words, we could extract the average word, syllable, and character rate (*Figure 2—figure supplements 2* and *3*). The reported rates fall within the generally reported ranges for syllables (5.2 Hz) and words (3.7 Hz; *Ding et al., 2017*; *Pellegrino and Coupé, 2011*).

We predict that knowledge about the language statistics influences the duration of speech units. As such we predict that more prevalent words will have on average a shorter duration (also reported in *Monsell et al., 1989*). In *Figure 2A*, the duration of several mono- and bi-syllabic words are listed with their word frequency. From these examples, it seems that words with higher word frequency

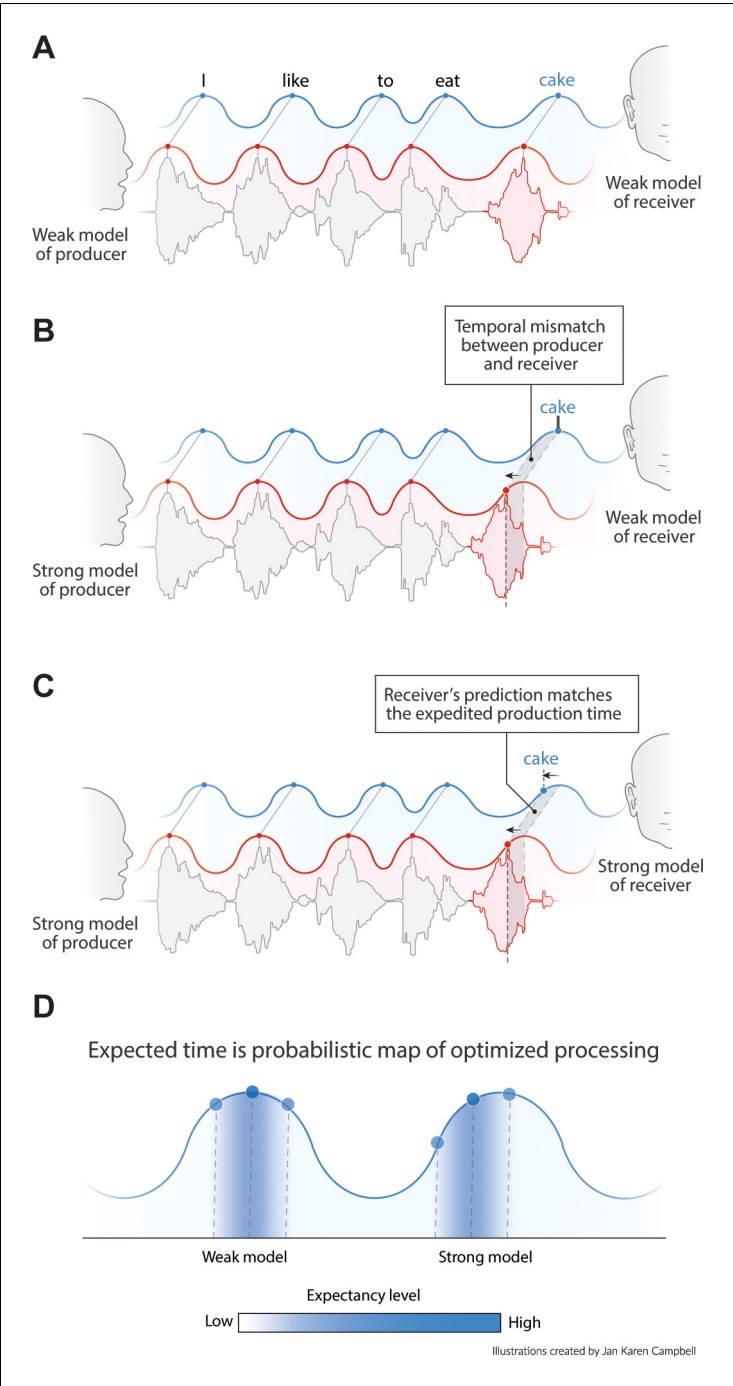

**Figure 1.** Proposed interaction between speech timing and internal linguistic models. (**A**) Isochronous production and expectation when there is a weak internal model (even distribution of node activation). All speech units arrive around the most excitable phase. (**B**) When the internal model of the producer does not align with the model of the receiver temporal alignment and optimal communication fails. (**C**) When both producer and receiver have a strong internal model, speech is non-isochronous and not aligned to the most excitable phase, but fully expected by the brain. (**D**) Expected time is a constraint distribution in which the center can be shifted due to linguistic constraints.

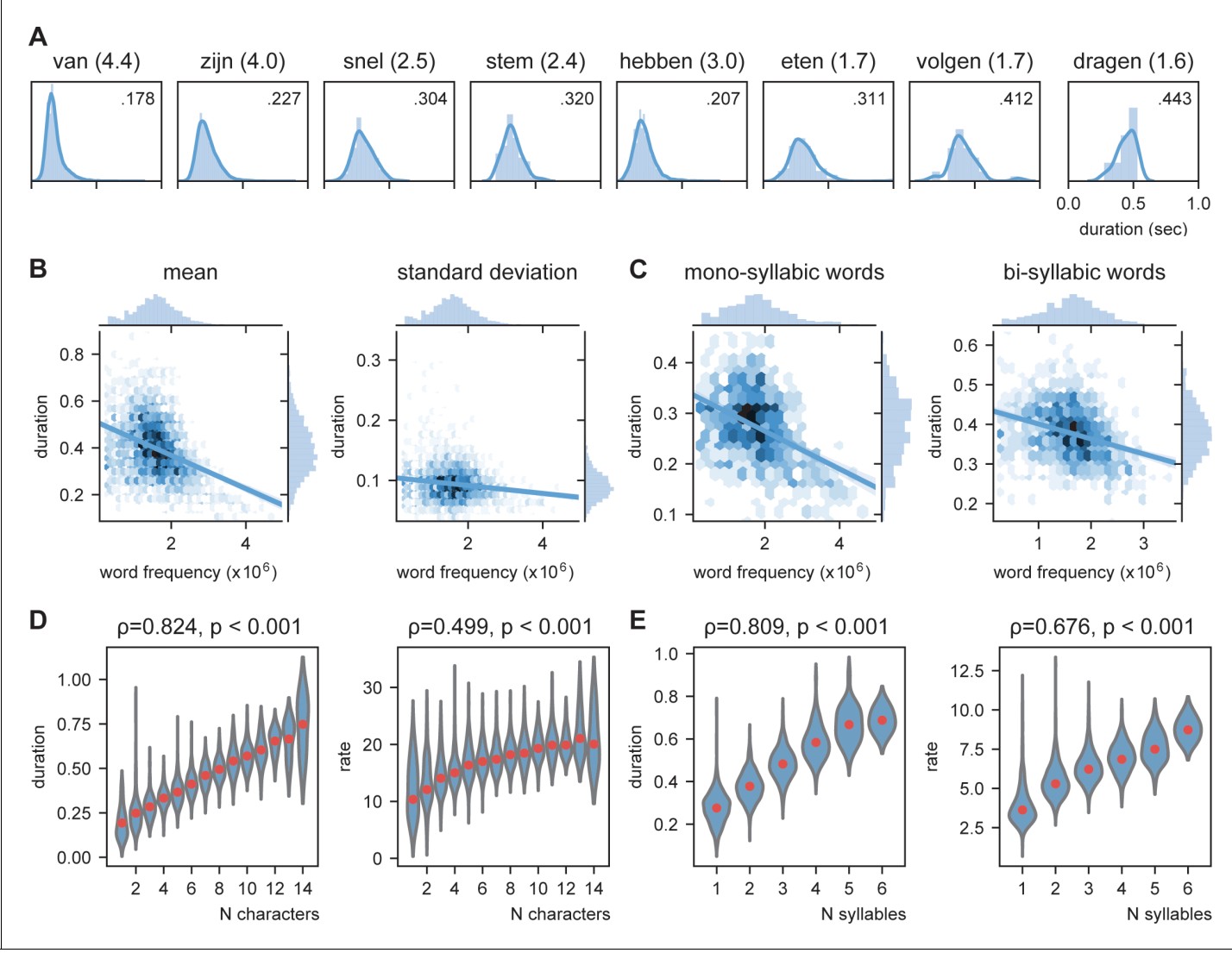

**Figure 2.** Word frequency modulates word duration. (A) Example of mono- and bi-syllabic words of different word frequencies in brackets (van=from, zijn=be, snel=fast, stem=voice, hebben=have, eten=eating, volgend=next, toekomst=future). Text in the graph indicates the mean word duration. (B) Relationship between word frequency and duration. Darker colors mean more values. (C) same as (B) but separately for mono- and bi-syllabic words. (D) Relationship character amount and word duration. The longer the words, the longer the duration (left). The increase in word duration does not follow a fixed number per character as duration as measured by rate increases. (E) same as (D) but for number of syllables. Red dots indicate the mean.

The online version of this article includes the following figure supplement(s) for figure 2:

**Figure supplement 1.** Distribution of mean duration (A) and of average rate (B).

**Figure supplement 2.** Distribution of mean duration split up for word length (in characters).

**Figure supplement 3.** Distribution of mean duration split up for syllable length.

generally have a shorter duration. To test this statistically we entered word frequency in an ordinary least square regression with number of syllables as control. Both number of syllables (coefficient = 0.1008, t(2843) = 75.47, p<0.001) as well as word frequency (coefficient = −0.022, t(2843) = −13.94, p<0.001) significantly influence the duration of the word. Adding an interaction term did not significantly improve the model (F (1,2843) = 1.320, p=0.251; *Figure 2B,C*). The effect is so strong that words with a low frequency can last three times as long as high-frequency words (even within mono-syllabic words). This indicates that word frequency could be an important part of an internal model that influences word duration.

The previous analysis probed us to expand on the relationship between word duration and length of the words. Obviously, there is a strong correlation between word length and mean word duration

(number of characters 0.824, p<0.001; number of syllables: ρ = 0.808, p<0.001; for number of syllables already shown above; *Figure 2D,E*). In contrast, this correlation is present, but much lower for the standard deviation of word duration (number of characters: ρ = 0.269, p<0.001; number of syllables: ρ = 0.292, p<0.001). Finding a strong correlation does not imply that for every time unit increase in the word length, the duration of the word also increases with the same time unit, i.e., bi-syllabic words do not necessarily have to last twice as long as mono-syllabic words. Therefore, we recalculated word duration to a rate unit considering the number of syllables/characters of the word. Thus, a 250 ms mono- versus bi-syllabic word would have a rate of 4 versus 8 Hz, respectively. Then we correlated character/syllabic rate with word duration. If word duration increases monotonically with character/syllable length, there should be no correlation. We found that the syllabic rate varies between 3 and 8 Hz as previously reported (*Figure 2E*, right; *Ding et al., 2017*; *Pellegrino and Coupé, 2011*). However, the more syllables there are in a word, the higher this rate (ρ = 0.676, p<0.001). This increase was less strong for the character rate (ρ = 0.499, p<0.001; *Figure 2D*, right).

These results show that the syllabic/character rate depends on the number of characters /syllables within a word and is not an independent temporal unit (*Ghitza, 2013*). This effect is easy to explain when assuming that the prediction strength of an internal model influences word duration: transitional probabilities of syllables are simply more constrained within a word than across words (*Thompson and Newport, 2007*). This will reduce the time it takes to utter/perceive any syllable which is later in a word. In the current model, we focus on words (based on the availability of word2vec embedding used to calculate contextual predictabilities based on a RNN) instead of syllables, so we will not test this prediction for syllables, but instead we can investigate the effect of transitional probabilities and other statistical regularities flowing from internal models across words (see next section and [*Jadoul et al., 2016*] for statistical regularities in syllabic processing).

## Word-by-word predictability predicts word onset differences

The brain's internal model likely provides predictions about what linguistic features and representations, and possibly about which specific units, such as words, to expect next when listening to ongoing speech (*Martin, 2016*; *Martin, 2020*). As such, it is also expected that word-by-word onset delays are shorter for words that fit the internal model (i.e., those that are expected; *Beattie and Butterworth, 1979*). To investigate this possibility, we created a simplified version of an internal model predicting the next word using recurrent neural nets (RNN). We trained an RNN to predict the next word from ongoing sentences (*Figure 3A*). The model consisted of an embedding layer (pretrained; github.com/coosto/dutch-word-embeddings), a recurrent layer with a tanh activation function, and a dense output layer with a softmax activation. To prevent overfitting, we added a 0.2 dropout to the recurrent layers and the output layer. An Adam optimizer was used at a 0.001 learning rate and a batch size of 32. We investigated four different recurrent layers (GRU and LSTM at either 128 or 300 units; see *Figure 3—figure supplement 1*). The final model we use here includes a LSTM with 300 units. Input data consistent of 10 sequential words (label encoding) within the corpus (of a single speaker; shifting the sentences by one word at a time), and an output consisted of a single word. A maximum of four unknown labeled words (words not included in the word2vec estimations. Four was chosen as it was <50% of the words) was allowed in the input, but not in output. Validation consisted of a randomly chosen 2% of the data.

The output of the RNN reflects a probability distribution in which the values of the RNN sum up to one and each word has its own predicted value (*Figure 3A*; see *Figure 3—figure supplement 2* for differences across words and sentence position). As such we can extract the predicted value of the uttered word and relate the RNN prediction with the stimulus onset delay relative to the previous word. We entered word prediction in a regression using the stimulus onset difference between the current word in the sentence and the previous word (i.e., onset difference of words). We added the control variables bigram (using the NLTK toolbox based on the training data only), frequency of previous word, syllable rate (rate of the full sentence input), and mean duration of previous word (all variables that can account for part of the variance that affects the duration of the last word). We only used the test data (total of 7361 sentences, excluding all words in which the previous word (W-1) was not present in Celex. 4837 sentences). Many of the variables were skewed to the right; therefore, we transformed the data accordingly (see *Table 1*; results were robust to changes in these transformation).

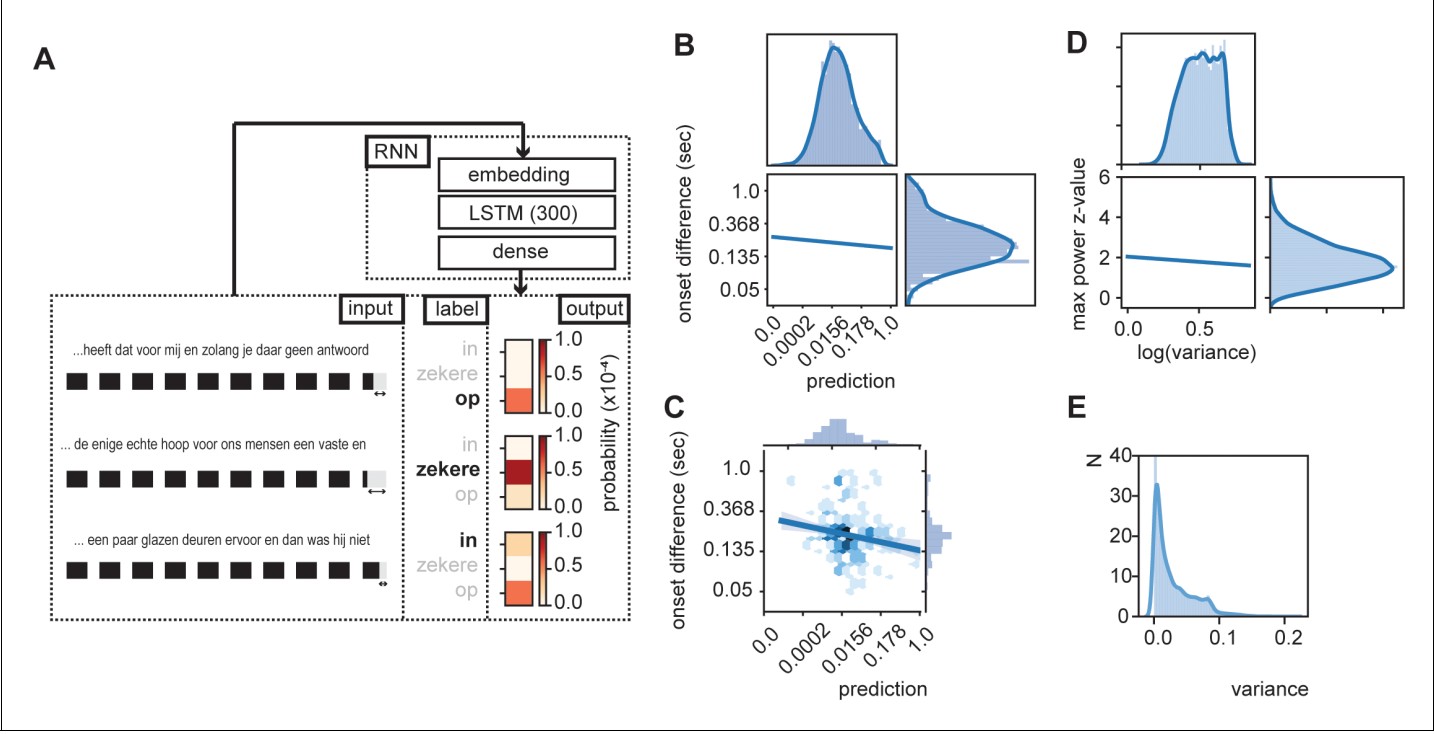

**Figure 3.** RNN output influence word onset differences. (A) Sequences of 10 words were entered in an RNN in order to predict the content of the next word. Three examples are provided of input data with the label (bold word) and probability output for three different words. The regression model showed a relation between the duration of last word in the sequence and the predictability of the next word such that words were systematically shorter when the next word was more predictable according to the RNN output (illustrated here with the shorted black boxes). (B) Regression line estimated at mean value of word duration and bigram. (C) Scatterplot of prediction and onset difference of data within ± 0.5 standard deviation of word duration and bigram. Note that for (B) and (C), the axes are linear on the transformed values. (D) Regression line for the correlation between logarithm of variance of the prediction and theta power. (E) None-transformed distribution of variance of the predictions (within a sentence). Translation of the sentences in (A) from top to bottom: '... that it has for me and while you have no answer [on]', '... the only real hope for us humans is a firm and [sure]', '... a couple of glass doors in front and then it would not have been [in]'.

The online version of this article includes the following figure supplement(s) for figure 3:

**Figure supplement 1.** Recurrent neural network evaluation.

**Figure supplement 2.** RNN prediction distributions.

All predictors except word frequency of the previous word showed a significant effect (**Table 1**). The variance explained by word frequency was likely captured by the mean duration variable of the previous word, which is correlated to word frequency. The RNN predictor could capture more variance than the bigram model, suggesting that word duration is modulated by the level of predictability within a fuller context than just the conditional probability of the current word given the previous

**Table 1.** Summary of regression model for logarithm of onset difference of words.

| Variable | Trans | B | β | SE | t | p | VIF |
|---|---|---|---|---|---|---|---|
| Intercept | x | 0.9719 | | 0.049 | 19.764 | <0.001 | |
| RNN prediction | x $^{(1/6)}$ | −0.3370 | −0.0862 | 0.047 | −7.163 | <0.001 | 1.5 |
| Bigram | log(x) | −0.0118 | −0.0316 | 0.005 | −2.424 | 0.015 | 1.8 |
| Word frequency W-1 | x | 0.0049 | 0.0076 | 0.009 | 0.546 | 0.585 | 2.0 |
| Mean duration W-1 | log(x) | 1.1206 | 0.7003 | 0.022 | 50.326 | <0.001 | 2.0 |
| Syllable Rate | x | −0.1033 | −0.2245 | 0.004 | −23.014 | <0.001 | 1.0 |

Model $R^2$ = 0.542. Trans = transformation, W-1 = previous word, B = unstandardized coefficient, β = standardized coefficient, SE = standard error, t = t value, p = p value, VIF = variance inflation factor.

word (*Figure 3B,C*). Importantly, it was necessary to use the trained RNN model as a predictor; entering the RNN predictions after the first training cycle (of a total of 100) did not results in a significant predictor (t(4837) = −1.191, p=0.234). Also adding the predictor word frequency of the current word did not add significant information to the model (F(1, 4830) = 0.2048, p=0.651). These results suggest that words are systematically lengthened (or pauses are added. However, the same predictors are also significant when excluding sentences containing pauses) when the next word is not strongly predicted by the internal model. We also investigate whether RNN predictions have an influence on the duration of the word that has to be uttered. We found no effect on the duration (Supporting *Table 1*).

## Sentence isochrony depends on prediction variance

In the previous section, we investigated word-to-word onsets, but did not investigate how this influences the temporal properties within a full sentence. In a regular sentence, predictability values change from word-to-word. Based on the previous results, it is expected that overall sentences with a more stable predictability level (sequential words are equally predictable) should be more isochronous than sentences in which the predictability shifts from high to low. This prediction is based on the observation that when predictions are equal the expected shift is the same, while for varying predictions, temporal shifts vary (*Figure 3B,C*).

To test this hypothesis, we extracted the RNN prediction for 10 subsequent words. Then we extracted the variance of the prediction across those 10 words and extracted the word onset itself. We created a time course at which word onset were set to 1 (at a sampling rate of 100 Hz). Then we performed an fast Fourier transform (FFT) and extracted z-transformed power values over a 0–15 Hz interval. The power at the maximum power value with the theta range (3–8 Hz) was extracted. These max z-scores were correlated with the log transform of the variance (to normalize the skewed variance distribution; *Figure 3E*). We found a weak, but significant negative correlation (r = −0.062, p<0.001; *Figure 3D*) in line with our hypothesis. This suggests that the more variable the predictions within a sentence, the lower the peak power value is. When we repeated the analysis on the envelope, we did not find a significant effect.

# Materials and methods

## Speech Tracking in a Model Constrained Oscillatory Network

In order to investigate how much of these duration effects can be explained using an oscillator model, we created the model Speech Tracking in a Model Constrained Oscillatory Network (STiMCON). STiMCON in its current form will not be exhaustive; however, it can extract how much an oscillating network can cope with asynchronies by using its own internal model illustrating how the brain's language model and speech timing interact (*Guest and Martin, 2021*). The current model is capable of explaining how top-down predictions can influence the processing time as well as provide an explanation for two known temporal illusions in speech.

STiMCON consists of a network of semantic nodes of which the activation A of each level in the model l is governed by:

$$A_{l,T} = C_{l-1 \to l} * A_{l-1,T} + C_{l+1 \to l} * A_{l+1,T} + inhib(Ta) + osc(T) \tag{1}$$

in which C represents the connectivity patterns between different hierarchical levels, T the time in a sentence, and Ta the vector of times of an individual node in an inhibition function (in milliseconds). The inhibition function is a gate function:

$$inhib(Ta) = \begin{cases} -3 * BaseInhib, Ta \\ 3 * BaseInhib, 20 \leq Ta < 100 \\ BaseInhib, Ta > 100 \end{cases} \tag{2}$$

in which BaseInhib is a constant for the base level of inhibition (negative value, set to −0.2). As such nodes are by default inhibited, as soon as they get activated above threshold (activation threshold set at 1) Ta sets to zero. Then, the node will have suprathreshold activation, which after 20 ms returns to increased inhibition until the base level of inhibition is returned. These values are set to

reflect early excitation and longer lasting inhibition, which are only loosely related to neurophysiological time scales. The oscillation is a constant oscillator:

$$osc(T) = Am * e^{2\pi i\omega T + i\varphi} \tag{3}$$

in which Am is the amplitude of the oscillator, ω the frequency, and φ the phase offset. As such we assume a stable oscillator which is already aligned to the average speech rate (see *Rimmele et al., 2018*; *Poeppel and Assaneo, 2020* for phase alignment models). The model used for the current simulation has one an input layer (l−1 level) and one single layer of semantic word nodes (l level) that receives feedback from a higher level layer (l+1 level). As such only the word (l) level is modeled according to *Equation 1–3* and the other levels form fixed input and feedback connection patterns. Even though the feedback influences the activity at the word level, it does not cause a phase reset as the phase of the oscillation does not change in response to this feedback.

## Language models influence time of activation

To illustrate how STiMCON can explain how processing time depends on the prediction of internal language models, we instantiated a language model that had only seen three sentences and five words presented at different probabilities (I eat cake at 0.5 probability, I eat nice cake at 0.3 probability, I eat very nice cake at 0.2 probability; *Table 2*). While in the brain the prediction should add up to 1, we can assume that the probability is spread across a big number of word nodes of the full language model and therefore neglectable. This language model will serve as the feedback arriving from the l+1-level to the l-level. The l-level consists of five nodes that each represent one of the words and receives proportional feedback from l+1 according to *Table 2* with a delay of 0.9*ω seconds, which then decays at 0.01 unit per millisecond and influences the l-level at a proportion of 1.5. The 0.9*ω was defined as we hypothesized that onset time would be loosely predicted around on oscillatory cycle, but to be prepared for input slightly earlier (which of course happens for predictable stimuli), we set it to 0.9 times the length of the cycle. The decay is needed and set such that the feedback would continue around a full theta cycle. The proportion was set empirically such to ensure that strong feedback did cause suprathreshold activation at the active node. The feedback is only initiated when supra-activation arrives due to l−1-level bottom-up input. Each word at the l−1-level input is modeled as a linearly function to the individual nodes lasting length of 125 ms (half a cycle, ranging from 0 to 1 arbitrary units). As such, the input is not the acoustic input itself but rather reflects a linear increase representing the increasing confidence of a word representing the specific node. φ is set such that the peak of a 4 Hz oscillation aligns to the peak of sensory input of the first word. Sensory input is presented at a base stimulus onset asynchrony of 250 ms (i.e., 4 Hz).

When we present this model with different sensory inputs at an isochronous rhythm of 4 Hz, it is evident that the timing at which different nodes reach activation depends on the level of feedback that is provided (*Figure 4*). For example, while the /I/-node needs a while to get activated after the initial sensory input, the /eat/-node is activated earlier as it is pre-activated due to feedback. After presenting /eat/, the feedback arrives at three different nodes and the activation timing depends on the stimulus that is presented (earlier activation for /cake/ compared to /very/).

**Table 2.** Example of a language model.
This model has seen three sentences at different probabilities. Rows represent the prediction for the next word, e.g., /I/ predicts /eat/ at a probability of 1, but after /eat/ there is a wider distribution.

|      | I | Eat | Very | Nice | Cake |
|------|---|-----|------|------|------|
| I    | 0 | 1   | 0    | 0    | 0    |
| eat  | 0 | 0   | 0.2  | 0.3  | 0.5  |
| very | 0 | 0   | 0    | 1    | 0    |
| nice | 0 | 0   | 0    | 0    | 1    |
| cake | 0 | 0   | 0    | 0    | 0    |

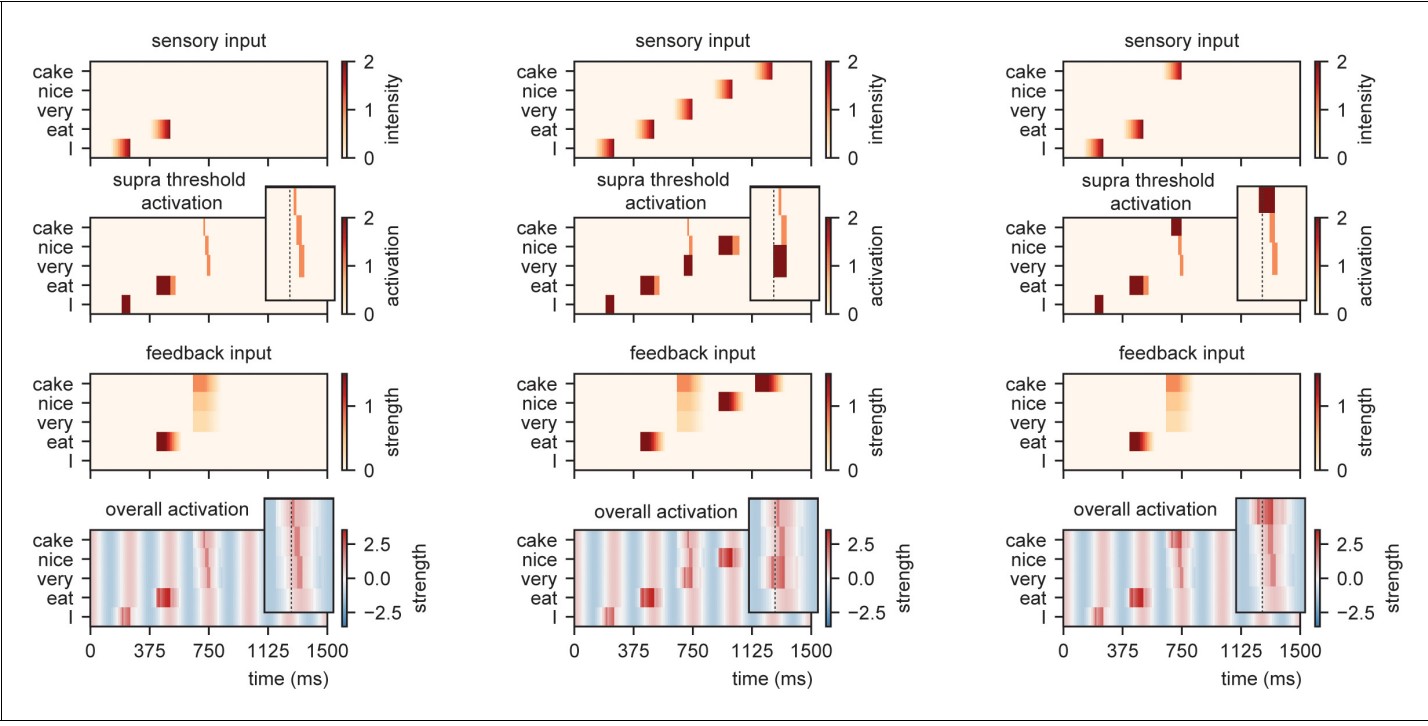

**Figure 4.** Model output for different sentences. For the supra-threshold activation dark red indicates activation which included input from l+1 as well as l1, orange indicates activation due to l+1 input. Feedback at different strengths causes phase dependent activation (left). Suprathreshold activation is reached earlier when a highly predicted stimulus (right) arrives, compared to a mid-level predicted stimulus (middle).

## Time of presentation influences processing efficiency

To investigate how the time of presentation influences the processing efficiency, we presented the model with /I eat XXX/ in which the last word was varied in content (*Figure 5A*; either /I/, /very/, /nice/, or /cake/), intensity (linearly ranging from 0 to 1), and onset delay (ranging between −125 and +125 ms relative to isochronous presentation). We extracted the time at which the node matching the stimulus presentation reached activation threshold first (relative to stimulus onset and relative to isochronous presentation).

*Figure 5B* shows the output. When there is no feedback (i.e., at the first word /I/ presentation), a classical efficiency map can be found in which processing is most optimal (possible at lowest stimulus intensities) at isochronous (in phase with the stimulus rate) presentation and then drops to either side. For nodes that have feedback, input processing is possible at earlier times relative to isochronous presentation and parametrically varies with prediction strength (earlier for /cake/ at 0.5 probability, then /very/ at 0.2 probability). Additionally, the activation function is asymmetric. This is a consequence of the interaction between the supra-activation caused by the feedback and the sensory input. As soon as supra-activation is reached due to the feedback, sensory input at any intensity will reach supra-activity (thus at early stages of the linearly increasing confidence of the input). This is why for the /very/ stimulus activation is still reached at later delays compared to /nice/ and /cake/ as the /very/-node reaches supra-activation due to feedback at a later time point. In regular circumstances, we would of course always want to process speech, also when it arrives at a less excitable phase. Note, however, that the current stimulus intensities were picked to exactly extract the threshold responses. When we increase our intensity range above 2.1, nodes will always get activated even on the lowest excitable phase of the oscillation.

When we investigate timing differences in stimulus presentation, it is important to also consider what this means for the timing in the brain. Before, we showed that the amount of prediction can influence timing in our model. It is also evident that the earlier a stimulus was presented the more time it took (relative to the stimulus) for the nodes to reach threshold (more yellow colors for earlier delays). This is a consequence of the oscillation still being at a relatively low excitability point at

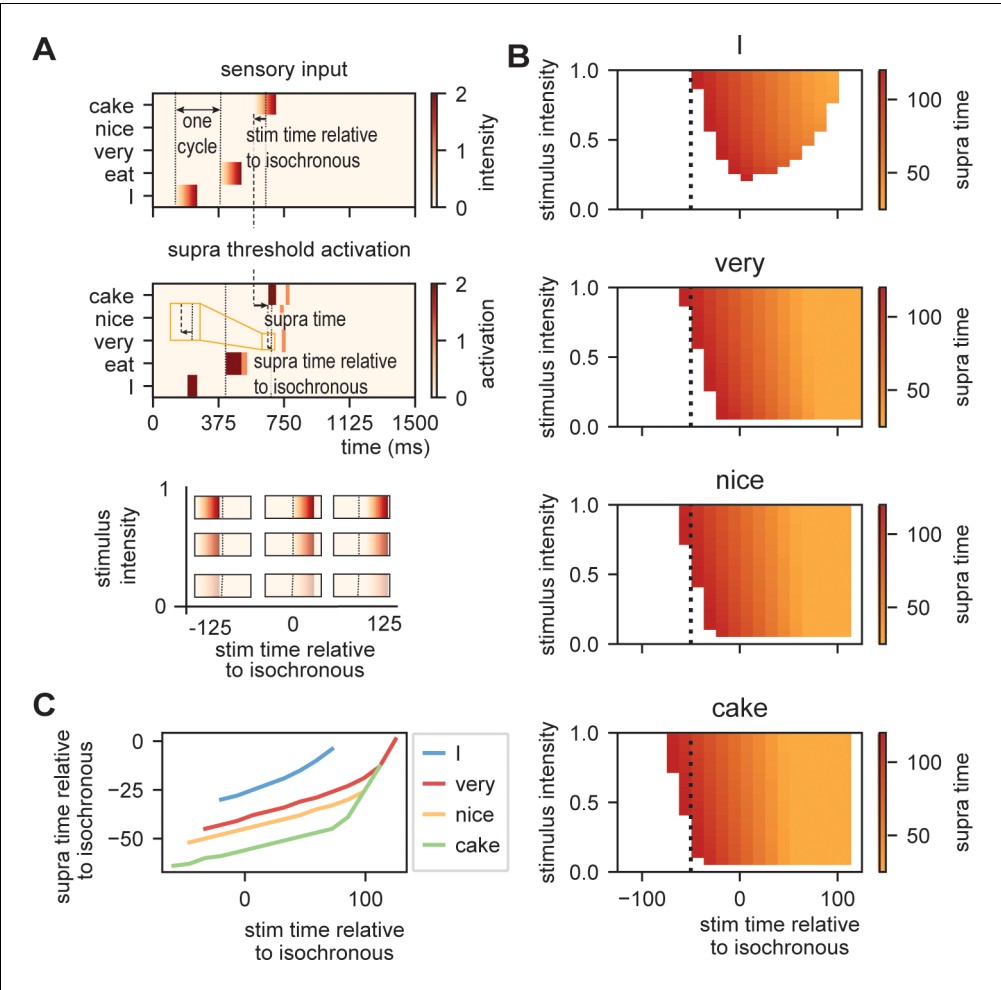

**Figure 5.** Model output on processing efficiency. (**A**) Input given to the model. Sensory input is varied in intensity and timing. We extract the time of activation relative to stimulus onset (supra-time) and relative to isochrony onset. (**B**) Time of presentation influences efficiency. Outcome variable is the time at which the node reached threshold activation (supra-time). The dashed line is presented to ease comparison between the four content types. White indicates that threshold is never reached. (C) Same as (B), but estimated at a threshold of 0.53 showing that oscillations regulate feedforward timing. Panel (C) shows that the earlier the stimuli are presented (on a weaker point of the ongoing oscillation), the longer it takes until supra-threshold activation is reached. This figure shows that timing relative to the ongoing oscillation is regulated such that the stimulus activation timing is closer to isochronous. Line discontinuities are a consequence of stimuli never reaching threshold for a specific node.

stimulus onset for stimuli that are presented early during the cycle. However, when we translate these activation threshold timing to the timing of the ongoing oscillation, the variation is strongly reduced (*Figure 5C*). A stimulus timing that varies between 130 ms (e.g., from −59 to +72 in the /cake/ line; excluding the non-linear section of the line) only reaches the first supra-threshold response with 19 ms variation in the model (translating to a reduction of 53–8% of the cycle of the ongoing oscillation, i.e., a 1:6.9 ratio). This means that within this model (and any oscillating model) the activation of nodes is robust to some timing variation in the environment. This effect seemed weaker when no prediction was present (for the /I/ stimulus this ratio was around 1:3.5. Note that when determining the /cake/ range using the full line the ratio would be 1:3.4).

## Top-down interactions can provide rhythmic processing for non-isochronous stimulus input

The previous simulation demonstrate that oscillations provide a temporal filter and the processing at the word layer can actually be closer to isochronous than what can be solely extracted from the stimulus input. Next, we investigated whether dependent on changes in top-down prediction, processing within the model will be more or less rhythmic. To do this, we create stimulus input of 10 sequential words at a base rate of 4 Hz to the model with constant (*Figure 6A*; low at 0 and high at 0.8 predictability) or alternating word-to-word predictability. For the alternating conditions, word-to-word predictability alternates between low and high (sequences which word are predicted at 0 or 0.8 predictability, respectively) or shift from high to low. For this simulation, we used Gaussian sensory input (with a standard deviation of 42 ms aligning the mean at the peak of the ongoing oscillation; see *Figure 6—figure supplement 1* for output with linear sensory input). Then, we vary the onset time of the odd words in the sequence (shifting from −100 up to +100 ms) and the stimulus intensity (from 0.2 to 1.5). We extracted the overall activity of the model and computed the FFT of the created time course (using a Hanning taper only including data from 0.5 to 2.5 s to exclude the onset responses). From this FFT, we extracted the peak activation at the stimulation rate of 4 Hz.

The first thing that is evident is that the model with no content predictions has overall lowest power, but has the strongest 4 Hz response around isochronous presentation (odd word offset of 0 ms) at high stimulus intensities (*Figure 6B–D*) following closely the acoustic input. Adding overall high predictability increases the power, but also here the power seems symmetric around zero. The spectra of the alternating predictability conditions look different. For the low to high predictability condition, the curve seems to be shifted to the left such that 4 Hz power is strongest when the predictable odd stimulus is shifted to an earlier time point (low–high condition). This is reversed for the high–low condition. At middle stimulus intensities, there is a specific temporal specificity window at which the 4 Hz power is particularly strong. This window is earlier for the low–high than the high–low alternation (*Figure 6C,D* and *Figure 6—figure supplement 2*). The effect only occurs at specific middle-intensity combination as at high intensities the stimulus dominates the responses and at low intensities the stimulus does not reach threshold activation. These results show that even though stimulus input is non-isochronous, the interaction with the internal model can still create a potential isochronous structure in the brain (see *Meyer et al., 2019*; *Meyer et al., 2020*). Note that the direction in which the brain response is more isochronous matches with the natural onset delays in speech (shorter onset delays for more predictable stimuli).

## Model validation

### STiMCON's sinusoidal modulations of RNN predictions is optimally sensitive to natural onset delays

Next, we aimed to investigated whether STiMCON would be optimally sensitive to speech input timings found naturally in speech. Therefore, we tried to fit STIMCON's expected word-to-word onset differences to the word-to-word onset differences we found in the CGN. At a stable level of intensity of the input and inhibition, the only aspect that changes the timing of the interaction between top-down predictions and bottom-up input within STiMCON is the ongoing oscillation. Considering that we only want to model for individual words how much the prediction $(C_{l+1 \to l} * A_{l+1,T})$ influences the expected timing we can set the contribution of the other factors from *Equation (1)* to zero remaining with the relative contribution of prediction:

$$C_{l+1 \to l} * A_{l+1,T} = top\ down\ influence\ = -osc(T) \tag{4}$$

We can solve this formula in order to investigate the expected relative time shift (T) in processing that is a consequence of the strength of the prediction (ignoring that in the exact timing will also depend on the strength of the input and inhibition):

$$relative\ time\ shift = \frac{1}{2\pi\omega}\left(\arcsin\left(\frac{C_{l+1 \to l} * A_{l+1,T}}{-Am}\right) - \varphi\right) \tag{5}$$

ω was set as the syllable rate for each sentence, and Am and φ were systematically varied. We fitted a linear model between the STiMCON's expected time and the actual word-to-word onset

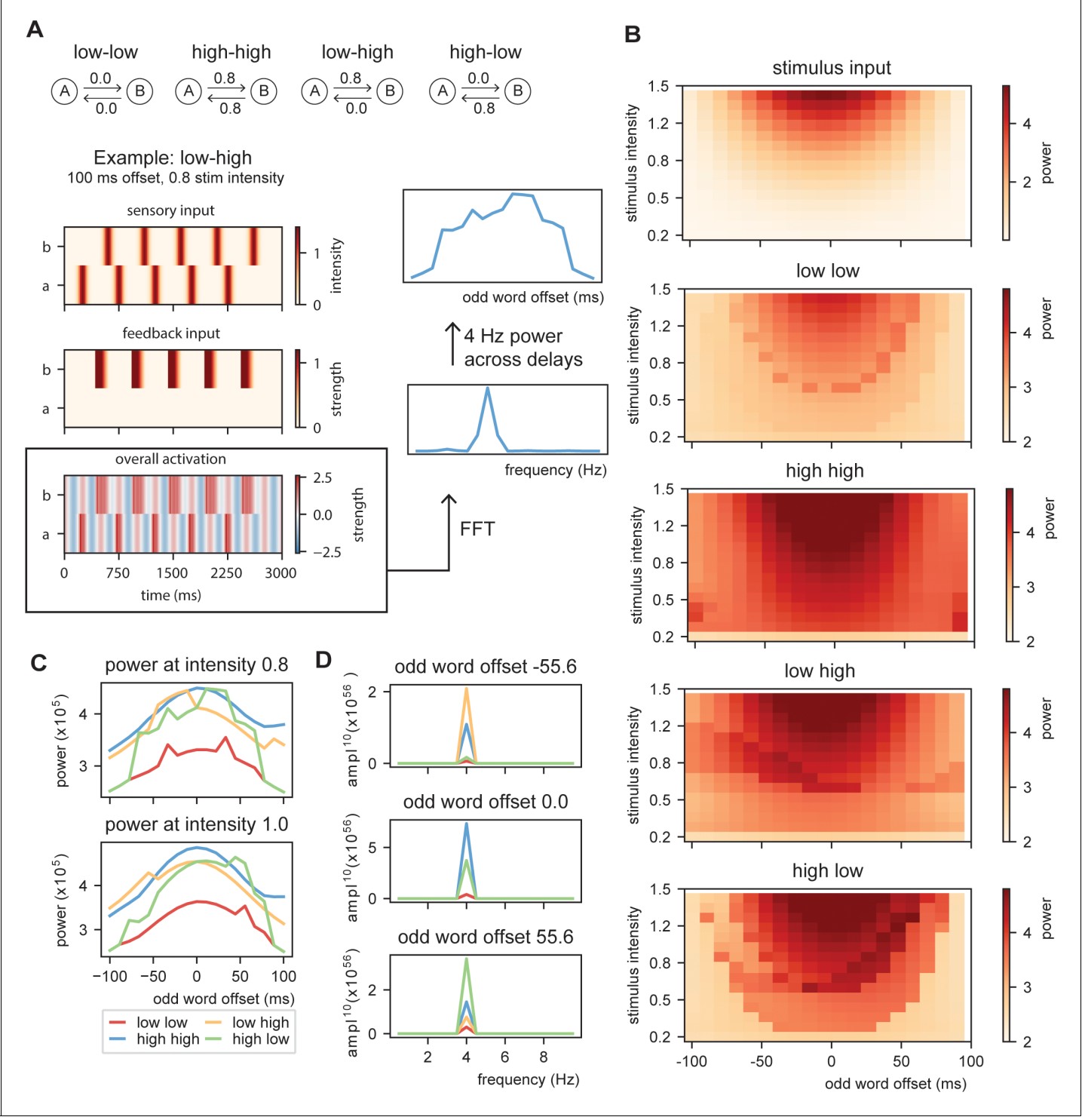

**Figure 6.** Model output on rhythmicity. (**A**) We presented the model with repeating (**A**, **B**) stimuli with varying internal models. We extracted the power spectra and peak activity at various odd stimulus offsets and stimulus intensities. (**B**) Strength of 4 Hz power depends on predictability in the stream. When predictability is alternated between low and high, activation is more rhythmic when the predictable odd stimulus arrives earlier and vice versa. (**C**) Power across different internal models at intensity of 0.8 and 1.0 (different visualization than **B**). (**D**) Magnitude spectra at three different odd word offsets at 1.0 intensity. To more clearly illustrate the differences, the magnitude to the power of 20 is plotted.

The online version of this article includes the following figure supplement(s) for figure 6:

**Figure supplement 1.** Power at 4 Hz using linearly increasing sensory input.

**Figure supplement 2.** Example of overall activation at threshold 0.8 (Gaussian shaped input).

differences. This model was similar to the model described in the section *Word-by-word predictability predicts word onset differences* and included the predictor syllable rate and duration of the previous word. However, as we were interested in how well non-transformed data matches the natural onset timings, we did not perform any normalization besides *Equation (5)*. As this might involve violating some of the assumptions of the ordinary least square fit, we estimate model performance by repeating the regression 1000 times fitting it on 90% of the data (only including the test data from the RNN) and extracting $R^2$ from the remaining 10%.

Results show a modulation of the $R^2$ dependent on the amplitude and phase offset of the oscillation (*Figure 7A*). This was stronger than a model in which transformation in *Equation (5)* was not applied ($R^2$ for a model with no transfomation was 0.389). This suggests that STiMCON expected time durations matches the actual word-by-word duration. This was even more strongly so for specific oscillatory alignments (around $-0.25\pi$ offset), suggesting an optimal alignment phase relative to the ongoing oscillation is needed for optimal tracking (*Giraud and Poeppel, 2012*; *Schroeder and Lakatos, 2009*). Interestingly, the optimal transformation seemed to automatically alter a highly skewed prediction distribution (*Figure 7B*) toward a more normal distribution of relative time shifts (*Figure 7C*). Note that the current prediction only operated on the word node (to which we have the RNN predictions), while full temporal shifts are probably better explained by word, syllabic, and phrasal predictions.

## STiMCON can explain perceptual effects in speech processing

Due to the differential feedback strength and the inhibition after suprathreshold feedback stimulation, STiMCON is more sensitive to lower predictable stimuli at phases later in the oscillatory cycle. This property can explain two illusions that have been reported in the literature, specifically, the observation that the interpretation of ambiguous input depends on the phase of presentation (*Ten Oever and Sack, 2015*; *Kayser et al., 2016*; *Ten Oever et al., 2020*) and on speech rate (*Bosker and Reinisch, 2015*). The only assumption that has to be made is that there is an uneven base prediction balance between the ways the ambiguous stimulus can be interpreted.

The empirical data we aim to model comprises an experiment in which ambiguous syllables, which could either be interpreted as /da/ or /ga/, were presented (*Ten Oever and Sack, 2015*). In

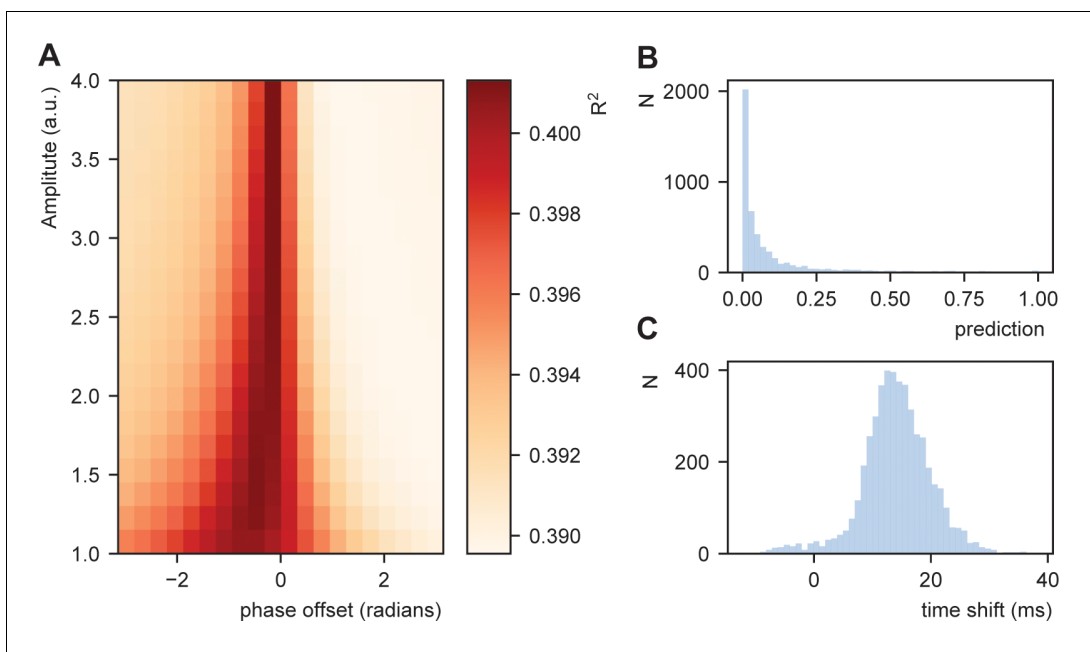

**Figure 7.** Fit between real and expected time shift dependent on predictability. (**A**) Phase offset and amplitude of the oscillation modulate the fit to the word-to-word onset durations. (**B**) Histogram of the predictions created by the deep neural net. (**C**) Histogram of the relative time shift transformation at phase of $-0.15\pi$ and amplitude of 1.5.

one of the experiments in this study, broadband simuli were presented at specific rates to entrain ongoing oscillations. After the last entrainment stimulus, an ambiguous /daga/ stimulus was presented at different delays (covering two cycles of the presentation rate at 12 different steps), putatively reflecting different oscillatory phases. Dependent on the delay of stimulation participants perceived either /da/ or /ga/, suggesting that phase modulates the percept of the participants. Besides this behavioral experiment, the authors also demonstrated that the same temporal dynamics were present when looking at ongoing EEG data, showing that the phase of ongoing oscillations at the onset of ambiguous stimulus presentation determined the percept (*Ten Oever and Sack, 2015*).

To illustrate that STiMCON is capable of showing a phase (or delay) dependent effect, we use an internal language model similar to our original model (*Table 2*). The model consists of four nodes (N1, N2, Nda, and Nga). N1 and N2 represent nodes responsive to two stimulus S1 and S2 that function as entrainment stimuli. N1 activation predicts a second unspecific stimulus (S2) represented by N2 at a predictability of 1. N2 activation predicts either da or ga at 0.2 and 0.1 probability, respectively. This uneven prediction of /da/ and /ga/ is justified as /da/ is more prevalent in the Dutch language as /ga/ (*Zuidema, 2010*), and it thus has a higher predicted level of occurring. Then, we present STiMCON (same parameters as before) with /S1 S2 XXX/. XXX is varied to have different proportion of the stimulus /da/ and /ga/ (ranging from 0% /da/ to 100% /ga/ in 12 times steps; these reflect relative proportions that sum up to one such that at 30% the intensity of /da/ would be at max 0.3 and of /ga/ 0.7) and is the onset is varied relate to the second to last word. We extract the time that a node reaches suprathreshold activity after stimulus onset. If both nodes were active at the same time, the node with the highest total activation was chosen. Results showed that for some ambiguous stimuli, the delay determines which node is activated first, modulating the ultimate percept of the participant (*Figure 8A*, also see *Figure 8—figure supplement 1A*). The same type of simulation can explain how speech rate can influence perception (*Figure 8—figure supplement 1B*; but see *Bosker and Kösem, 2017*).

To further scrutinize on this effect, we fitted our model to the behavioral data of *Ten Oever and Sack, 2015*. As we used an iterative approach in the simulations of the model, we optimized the model using a grid search. We varied the parameters of proportion of the stimulus being /da/ or /ga/ (ranging between 10:5:80%), the onset time of the feedback (0.1:0.1:1.0 cycle), the speed of the feedback decay (0:0.01:0.1), and a temporal offset of the final sound to account for the time it takes to interpret a specific ambiguous syllable (ranging between −0.05:0.01:0.05 s). Our first outcome variable was the node that show the first suprathreshold activation (Nda = 1, Nga = 0). If both nodes were active at the same time, the node with the highest total activates was chosen. If both nodes had equal activation or never reached threshold activation, we coded the outcome to 0.5 (i.e., fully ambiguous). These outcomes were fitted to the behavioral data of the 6.25 Hz and 10 Hz presentation rate (the two rates showing a significant modulation of the percept). This data was normalized to have a range between 0 and 1 to account for the model outcomes being binary (0, 0.5, or 1). As a second outcome measure, we also extracted the relative activity of the /da/ and /ga/ nodes by subtracting their activity and dividing by the summed activity. The activity was calculated as the average activity over a window of 500 ms after stimulus onset and the final time course was normalized between 0 and 1.

For the first node activation analysis, we found that our model could fit the data at an average explained variance of 43% (30% and 58% for 6.25 Hz and 10 Hz, respectively; *Figure 8C,D*). For the average activity analysis, we found a fit with 83% explained variance. Compared to the original sinus fit, this explained variance was higher for the average activation analysis (40% for three parameter sinus fit [amplitude, phase offset, and mean]). Note that for the first node activation analysis, our fit cannot account for variance ranging between 0–0.5 and 0.5–1, while the sinus fit can do this. If we correct for this (by setting the sinus fit to the closest 0, 0.5, or 1 value and doing a grid search to optimize the fitting), the average fit of the sinus is 21%. Comparing the fits of the rectified sinus versus the first node activation reveals an average Akaike information criterion of the model and sinus fits of −27.0 and −24.1, respectively. For the average activation analysis, this was −41.5 versus −27.8, respectively. This overall suggests that the STiMCON model has the better fit. Thus, STiMCON does better than a fixed-frequency sinus fit. This is a likely consequence of the sinus fit not being able to explain the dampening of the oscillation later (i.e., the perception bias is stronger for shorter compared to longer delays).

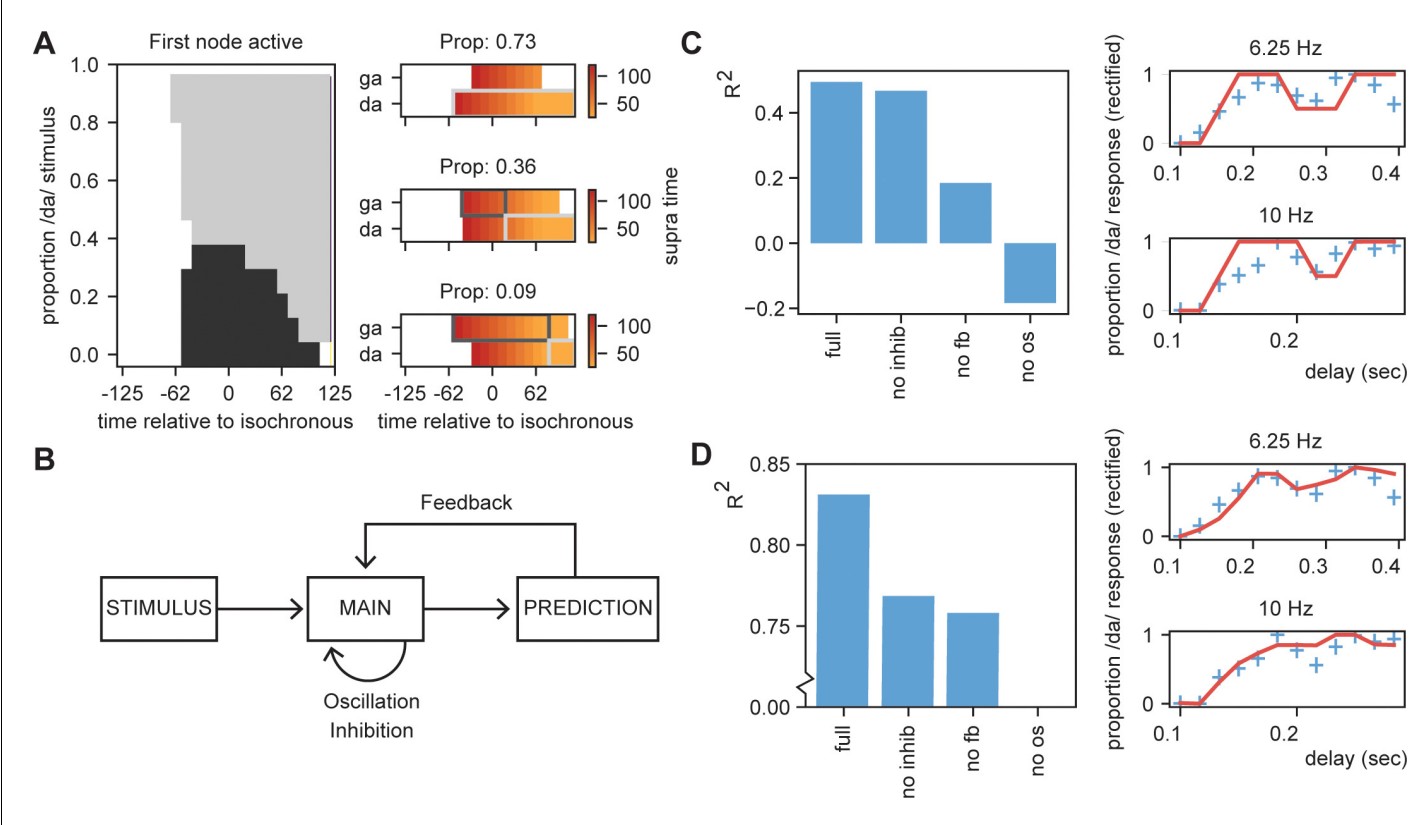

**Figure 8.** Results for /daga/ illusions. (**A**) Modulations due to ambiguous input at different times. Illustration of the node that is active first. Different proportions of the /da/ stimulus show activation timing modulations at different delays. (**B**) Summary of the model and the parameters altered for the empirical fits in (**C**) and (**D**). (**C**). R2 for the grid search fit of the full model using the first active node as outcome variable, a model without inhibition (no inhib), without uneven feedback (no fb), or without an oscillation (no os). The right panel shows the fit of the full model on the rectified behavioral data of *Ten Oever and Sack, 2015*. Blue crossed indicate rectified data and red lines indicate the fit. (**D**) is the same as (**C**) but using the average activity instead of the first active node. Removing the oscillation results in an $R^2$ less than 0.

The online version of this article includes the following figure supplement(s) for figure 8:

**Figure supplement 1.** Explaining speech timing illusions.

Finally, we investigated the relevance of the three key features of our model for this fit: inhibition, feedback, and oscillations (*Figure 8B*). We repeated the grid search fit but set either the inhibition to zero, the feedback matrix equal for both /da/ and /ga/ (both 0.15), or the oscillation at an amplitude of zero. Results showed for both outcome measures that the full model showed the best performance. Without the oscillation, the models could not even fit better than the mean of the model ($R^2 < 0$). Removing the feedback had a negative influence on both the outcome measures, dropping the performance. Removing the inhibition reduced performance for both outcome measures, but more strongly on the average activation compared to the first active node model. This suggest that all features (with potentially to a lesser extend the inhibition) are required to model the data, suggesting that oscillatory tracking is dependent on linguistic constrains flowing from the internal language model.

## Discussion

In the current paper, we combined an oscillating computational model with a proxy for linguistic knowledge, an internal language model, in order to investigate the model's processing capacity for onset timing differences in natural speech. We show that word-to-word speech onset differences in natural speech are indeed related to predictions flowing from the internal language model (estimated through an RNN). Fixed oscillations aligned to the mean speech rate are robust against

natural temporal variations and even optimized for temporal variations that match the predictions flowing from the internal model. Strikingly, when the pseudo-rhythmicity in speech matches the predictions of the internal model, responses were more rhythmic for matched pseudo-rhythmic compared to isochronous speech input. Our model is optimally sensitive to natural speech variations, can explain phase-dependent speech categorization behavior (*Ten Oever and Sack, 2015*; *Thézé et al., 2020*; *Reinisch and Sjerps, 2013*; *Ten Oever et al., 2020*), and naturally comprises a neural phase code (*Panzeri et al., 2015*; *Mehta et al., 2002*; *Lisman and Jensen, 2013*). These results show that part of the pseudo-rhythmicity of speech is expected by the brain and it is even optimized to process it in this manner, but only when it follows the internal model.

Speech timing is variable, and in order to understand how the brain tracks this pseudo-rhythmic signal, we need a better understanding of how this variability arises. Here, we isolated one of the components explaining speech time variation, namely, constraints that are posed by an internal language model. This goes beyond extracting the average speech rate (*Ding et al., 2017*; *Poeppel and Assaneo, 2020*; *Pellegrino and Coupé, 2011*) and might be key to understanding how a predictive brain uses temporal cues. We show that speech timing depends on the predictions made from an internal language model, even when those predictions are highly reduced to be as simple as word predictability. While syllables generally follow a theta rhythm, there is a systematic increase in syllabic rate as soon as more syllables are in a word. This is likely a consequence of the higher close probability of syllables within a word which reduces the onset differences of the later uttered syllables (*Thompson and Newport, 2007*). However, an oscillatory model constrained by an internal language model is sensitive to these temporal variations, it is actually capable of processing them optimally.

The oscillatory model we here pose has three components: oscillations, feedback, and inhibition. The oscillations allow for the parsing of speech and provide windows in which information is processed (*Giraud and Poeppel, 2012*; *Ghitza, 2012*; *Peelle and Davis, 2012*; *Martin and Doumas, 2017*). Importantly, the oscillation acts as a temporal filter, such that the activation time of any incoming signal will be confined to the high excitable window and thereby is relatively robust against small temporal variations (*Figure 5C*). The feedback allows for differential activation time dependent on the sensory input (*Figure 5B*). As a consequence, the model is more sensitive to higher predictable speech input and therefore active earlier on the duty cycle (this also means that oscillations are less robust against temporal variations when the feedback is very strong). The inhibition allows for the network to be more sensitive to less predictable speech units when they arrive later (the higher predictable nodes get inhibited at some point on the oscillation; best illustrated by the simulation in *Figure 8A*). In this way, speech is ordered along the duty cycle according to its predictability (*Lisman and Jensen, 2013*; *Jensen et al., 2012*). The feedback in combination with an oscillatory model can explain speech rate and phase-dependent content effects. Moreover, it is an automatic temporal code that can use time of activation as a cue for content (*Mehta et al., 2002*). Note that previously we have interpreted the /daga/ phase-dependent effect as a mapping of differences between natural audio-visual onset delays of the two syllabic types onto oscillatory phase (*Ten Oever et al., 2013*; *Ten Oever and Sack, 2015*). However, the current interpretation is not mutually exclusive with this delay-to-phase mapping as audio-visual delays could be bigger for less frequent syllables. The three components in the model are common brain mechanisms (*Malhotra et al., 2012*; *Mehta et al., 2002*; *Buzsáki and Draguhn, 2004*; *Bastos et al., 2012*; *Michalareas et al., 2016*; *Lisman, 2005*) and follow many previously proposed organization principles (e.g., temporal coding and parsing of information). While we implement these components on an abstract level (not veridical to the exact parameters of neuronal interactions), they illustrate how oscillations, feedback, and inhibition interact to optimize sensitivity to natural pseudo-rhythmic speech.

The current model is not exhaustive and does not provide a complete explanation of all the details of speech processing in the brain. For example, it is likely that the primary auditory cortex is still mostly modulated by the acoustic pseudo-rhythmic input and only later brain areas follow more closely the constraints posed by the language model of the brain. Moreover, we now focus on the word level, while many tracking studies have shown the importance of syllabic temporal structure (*Giraud and Poeppel, 2012*; *Ghitza, 2012*; *Luo and Poeppel, 2007*) as well as the role of higher order linguistic temporal dynamics (*Meyer et al., 2019*; *Kaufeld et al., 2020b*). It is likely that predictive mechanisms also operate on these higher linguistic levels as well as on syllabic levels. It is

known, for example, that syllables are shortened when the following syllabic content is known versus producing syllables in isolation (*Pluymaekers et al., 2005a*; *Lehiste, 1972*). Interactions also occur as syllables part of more frequent words are generally shortened (*Pluymaekers et al., 2005b*). Therefore, more hierarchical levels need to be added to the current model (but this is possible following *Equation (1)*). Moreover, the current model does not allow for phase or frequency shifts. This was intentional in order to investigate how much a fixed oscillator could explain. We show that onset times matching the predictions from the internal model can be explained by a fixed oscillator processing pseudo-rhythmic input. However, when the internal model and the onset timings do not match, the internal model phase and/or frequency shift are still required and need to be incorporated (see e.g. *Rimmele et al., 2018*; *Poeppel and Assaneo, 2020*).

We aimed to show that a stable oscillator can be sensitive to temporal pseudo-rhythmicities when these shifts match predictions from an internal linguistic model (causing higher sensitivity to these nodes). In this way, we show that temporal dynamics in speech and the brain cannot be isolated from processing the content of speech. This is in contrast with other models that try to explain how the brain deals with pseudo-rhythmicity in speech (*Giraud and Poeppel, 2012*; *Rimmele et al., 2018*; *Doelling et al., 2019*). While some of these models discuss that higher-level linguistic processing can modulate the timing of ongoing oscillations (*Rimmele et al., 2018*), they typically do not consider that in the speech signal itself the content or predictability of a word relates to the timing of this word. Phase resetting models typically deal with pseudo-rhythmicity by shifting the phase of ongoing oscillations in response to a word that is offset to the mean frequency of the input (*Giraud and Poeppel, 2012*; *Doelling et al., 2019*). We believe that this cannot explain how the brain uses what/when dependencies in the environment to infer the content of the word (e.g., later words are likely a less predictable word). Our current model does not have an explanation of how the brain can actually entrain to an average speech rate. This is much better described in dynamical systems theories in which this is a consequence of the coupling strength between internal oscillations and speech acoustics (*Doelling et al., 2019*; *Assaneo et al., 2021*). However, these models do not take top-down predictive processing into account. Therefore, the best way forward is likely to extend coupling between brain oscillations and speech acoustics (*Poeppel and Assaneo, 2020*), with the coupling of brain oscillations to brain activity patterns of internal models (*Cumin and Unsworth, 2007*).

In the current paper, we use an RNN to represent the internal model of the brain. However, it is unlikely that the RNN captures the wide complexities of the language model in the brain. The decades-long debates about the origin of a language model in the brain remains ongoing and controversial. Utilizing the RNN as a proxy for our internal language model makes a tacit assumption that language is fundamentally statistical or associative in nature, and does not posit the derivation or generation of knowledge of grammar from the input (*Chater, 2001*; *McClelland and Elman, 1986*). In contrast, our brain could as well store knowledge of language that functions as fundamental interpretation principles to guide our understanding of language input (*Martin, 2016*; *Martin, 2020*; *Hagoort, 2017*; *Martin and Doumas, 2017*; *Friederici, 2011*). Knowledge of language and linguistic structure could be acquired through an internal self-supervised comparison process extracted from environmental invariants and statistical regularities from the stimulus input (*Martin and Doumas, 2019*; *Doumas et al., 2008*; *Doumas and Martin, 2018*). Future research should investigate which language model can better account for the temporal variations found in speech.

A natural feature of our model is that time can act as a cue for content implemented as a phase code (*Lisman and Jensen, 2013*; *Jensen et al., 2012*). This code unravels as an ordered list of predictability strength of the internal model. This idea diverges from the idea that entrainment should align to the most excitable phase of the oscillation with the highest energy in the acoustics (*Giraud and Poeppel, 2012*; *Rimmele et al., 2018*). Instead, this type of phase coding could increase the brain representational space to separate information content (*Lisman and Jensen, 2013*; *Panzeri et al., 2001*). We predict that if speech nodes have a different base activity, ambiguous stimulus interpretation should dependent on the time/phase of presentation (see *Ten Oever and Sack, 2015*; *Ten Oever et al., 2020*). Indeed, we could model two temporal speech illusions (*Figure 8*, *Figure 8—figure supplement 1*). There have also been null results regarding the influence of phase on ambiguous stimulus interpretation (*Bosker and Kösem, 2017*; *Kösem et al., 2016*). For the speech rate effect, when modifying the time of presentation with a neutral entrainer (summed sinusoidals with random phase), no obvious phase effect was reported (*Bosker and*

*Kösem, 2017*). A second null result relates to a study where participants were specifically instructed to maintain a specific perception in different blocks which likely increases the pre-activation and thereby the phase (*Kösem et al., 2016*). Future studies need to investigate the use of temporal/phase codes to disambiguate speech input and specifically use predictions in their design.

The temporal dynamics of speech signals needs to be integrated with the temporal dynamics of brain signals. However, it is unnecessary (and unlikely) that the exact duration of speech matches with the exact duration of brain processes. Temporal expansion or squeezing of stimulus inputs occur regularly in the brain (*Eagleman et al., 2005*; *Pariyadath and Eagleman, 2007*), and this temporal morphing also maps to duration (*Eagleman, 2008*; *Terao et al., 2008*; *Ulrich et al., 2006*) or order illusions (*Vroomen and Keetels, 2010*). Our model predicts increased rhythmic responses for non-isochronous speech matching the internal model. The perceived rhythmicity of speech could therefore also be an illusion generated by a rhythmic brain signal somewhere in the brain.

When investigating the pseudo-rhythmicity in speech, it is important to identify situations where speech is actually more isochronous. Two examples are the production of lists (*Jefferson, 1990*) and infant-directed speech (*Fernald, 2000*). In both these examples, it is clear that a strong internal predictive language model is lacking either on the producer's or on the receiver's side, respectively. The infant-directed speech also illustrates that a producer might proactively adapt its speech rhythm to the expectations of the internal model of the receiver to align better with the predictions from the receiver's model (*Figure 9B*; similar to when you are speaking to somebody that is just learning a new language). Other examples in which speech is more isochronous is during poems, during emotional conversation (*Hawkins, 2014*), and in noisy situations (*Bosker and Cooke, 2018*). While speculative, it is conceivable that in these circumstances one puts more weight on a different level of hierarchy than the internal linguistic model. In the case of poems and emotional conversation, an emotional route might get more weight in processing. In the case of noisy situations, stimulus input has to pass the first hierarchical level of the primary auditory cortex which effectively gets more weight than the internal model.

## Conclusions

We argued that pseudo-rhythmicity in speech is in part a consequence of top-down predictions flowing from an internal model of language. This pseudo-rhythmicity is created by a speaker and expected by a receiver if they have overlapping internal language models. Oscillatory tracking of this signal does not need to be hampered by the pseudo-rhythmicity, but can use temporal variations as a cue to extract content information since the phase of activation parametrically relates to the likelihood of an input relative to the internal model. Brain responses can even be more isochronous to pseudo-rhythmic compared to isochronous speech if they follow the temporal delays imposed by the internal model. This account provides various testable predictions which, we list in *Table 3* and *Figure 9*. We believe that by integrating neuroscientific explanations of speech tracking with linguistic models of language processing (*Martin, 2016*; *Martin, 2020*), we can improve to explain temporal speech dynamics. This will ultimately aid our understanding of language in the brain and provide a means to improve temporal properties in speech synthesis.

## Code availability statement

Code for the creation of the main figures is available on GitHub (*Ten Oever & Martin, 2021*; copy archived at swh:1:rev:873a2bf5c79fe2f828e72e14ef74db409d387854).

## Acknowledgements

AEM was supported by the Max Planck Research Group and Lise Meitner Research Group 'Language and Computation in Neural Systems' from the Max Planck Society, and by the Netherlands Organization for Scientific Research (grant 016.Vidi.188.029 to AEM). *Figure 1* and *9* were created in collaboration with scientific illustrator Jan-Karen Campbell (http://www.jankaren.com).

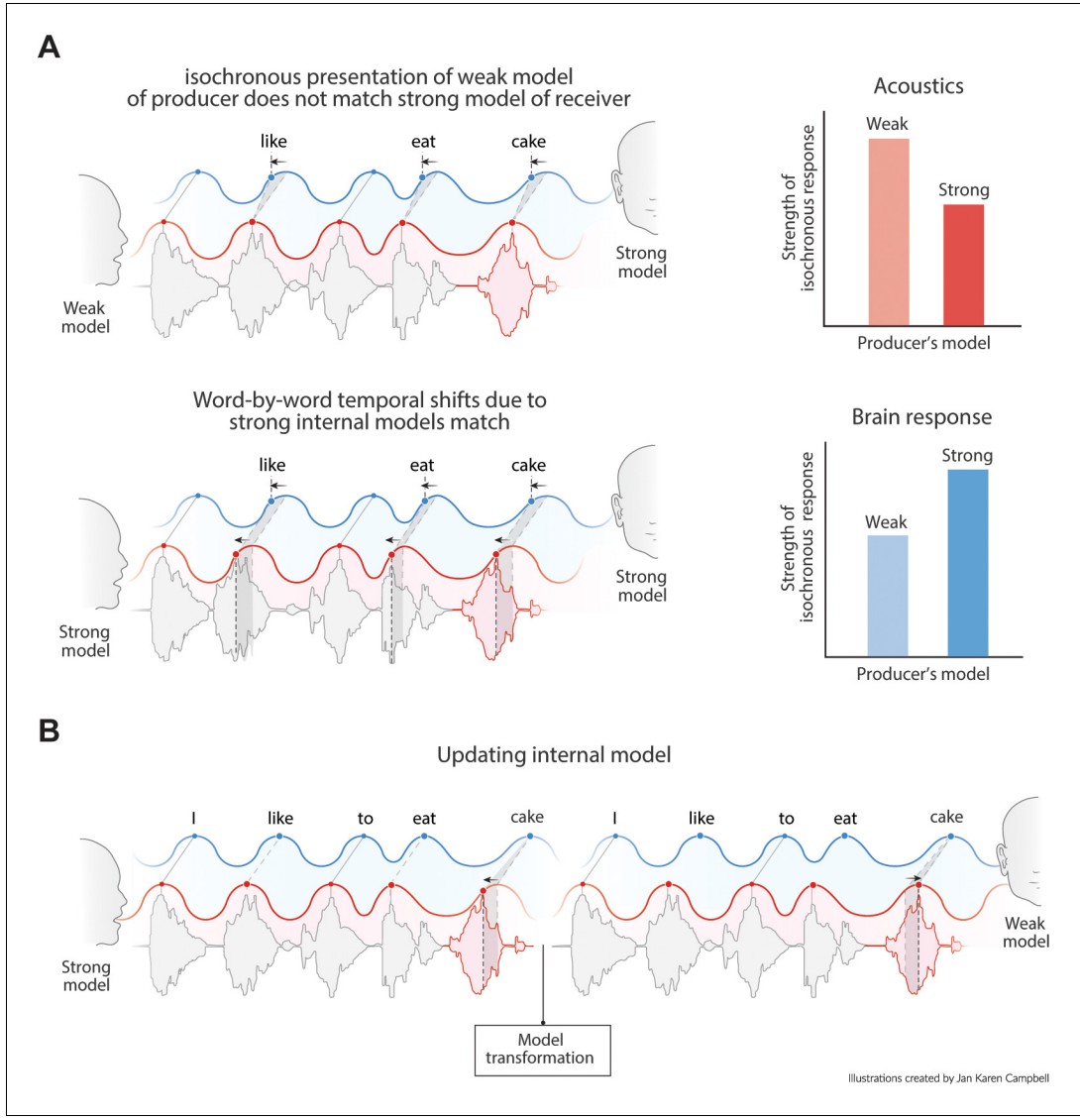

**Figure 9.** Predictions of the model. (**A**) Acoustics signals will be more isochronous when a producer has a weak versus a strong internal model (top right). When the producer's strong model matches the receiver's model, the brain response will be more isochronous for less isochronous acoustic input. (**B**) When a producer realizes the model of the receiver is weak, it might transform its model and thereby their speech timing to match the receiver's expectations.

**Table 3.** Predictions from the current model.

When there is a flat constraint distribution over an utterance (e.g., when probabilities are uniform over the utterance), the acoustics of speech should naturally be more isochronous (**Figures 9A** and **3D,E**).

If speech timing matches the internal language model, brain responses should be more isochronous even if the acoustics are not (**Figure 9A**).

The more similar the internal language models of two speakers, the more effective they are in 'entraining' each other's brain.

If speakers suspect their listener to have a flatter constraint distribution than themselves (e.g., the environment is noisy, or the speakers are in a second language context), they adjust to the distribution by speaking more isochronous (**Figure 9B**).

One adjusts the weight of the constraint distribution to a hierarchical level when needed. For example, when there is noise, participants adjust to the rhythm of primary auditory cortex instead of higher order language models. As a consequence, they speak more isochronous.

The theoretical account provides various predictions that are listed in this table.

## Additional information

### Funding

| Funder | Grant reference number | Author |
|---|---|---|
| Max Planck Society | MaxPlanck Research Group | Andrea E Martin |
| Nederlandse Organisatie voor Wetenschappelijk Onderzoek | 016.Vidi.188.029 | Andrea E Martin |
| Max Planck Society | Lise Meitner Research Group | Andrea E Martin |

The funders had no role in study design, data collection and interpretation, or the decision to submit the work for publication.

### Author contributions

Sanne ten Oever, Conceptualization, Data curation, Formal analysis, Visualization, Methodology, Writing - original draft, Writing - review and editing; Andrea E Martin, Conceptualization, Resources, Supervision, Funding acquisition, Validation, Writing - review and editing

### Author ORCIDs

Sanne ten Oever (iD) https://orcid.org/0000-0001-7547-5842
Andrea E Martin (iD) https://orcid.org/0000-0002-3395-7234

### Decision letter and Author response

Decision letter https://doi.org/10.7554/eLife.68066.sa1
Author response https://doi.org/10.7554/eLife.68066.sa2

## Additional files

### Supplementary files

- Supplementary file 1. Summary of regression model for logarithm of word duration.
- Transparent reporting form

### Data availability

Data used in the dataset relate to the corpus gesproken nederlands. Information about this dataset can be found here: http://lands.let.ru.nl/cgn/. Access to the dataset can be requested here: https://taalmaterialen.ivdnt.org/download/tstc-corpus-gesproken-nederlands/. Data regarding the simulations in Figure 8 are based on data from Ten Oever & Sack (2015). As this data regards a closed database owned by Maastricht University it is not openly available. However, the data is available upon request without any restrictions via sanne.tenoever@mpi.nl or datamanagement-fpn@maastrichtuniversity.nl.

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
