## [Decision Letter]

**Acceptance summary:**

The manuscript is of broad interest to readers in the field of speech recognition and neural oscillations. The authors provide a computational model which, in addition to feedforward acoustic input, incorporates linguistic predictions as feedback, allowing a fixed oscillator to process non-isochronous speech. The model is tested extensively by applying it to a linguistic corpus, EEG and behavioral data. The article gives new insights to the ongoing debate about the role of neural oscillations and predictability in speech recognition.

**Decision letter after peer review:**

Thank you for submitting your article "Oscillatory tracking of pseudo-rhythmic speech is constrained by linguistic predictions" for consideration by *eLife*. Your article has been reviewed by 3 peer reviewers, including Anne Kösem as the Reviewing Editor and Reviewer #1, and the evaluation has been overseen by Andrew King as the Senior Editor. The following individuals involved in review of your submission have agreed to reveal their identity: Johanna Rimmele (Reviewer #2); Keith Doelling (Reviewer #3).

Essential revisions:

All reviewers had a very positive assessment of the manuscript. They find the described work highly novel and interesting. However, they also find the manuscript quite dense and complex, and suggest some clarifications in the description of the model and in the methods.

Please find a list of the recommendations below.

*Reviewer #1 (Recommendations for the authors):*

1. First of all, I think that the concept of "internal language model" should be defined and described in more detail in the introduction. What does it mean when it is weak and when it is strong, for the producer and for the receiver?

2. It is still not fully clear to me what kind of information the internal oscillation is entraining to in the model. From Figure 1, it seems that the oscillation is driven by the acoustics only, but the phase of processing of linguistic units depends on their predictability.

3. If acoustic information arrives at the most excitable phase of the neural oscillation (as described in figure 1), and if predictability makes words arrive earlier, does it entail that more predictable words arrive at less excitable phases of the neural oscillation? What would be the computational advantage of this mechanism?

4. What is "stimulus intensity" in figure 5? Does it reflect volume or SNR?

5. Similarly what is "amplitude" in Figure 6?

6. L. 376-L. 439 "N2 activation predicts either da or ga at 0.2 and 0.1 probability respectively." Please explain why the probabilities are not equal in the model, same for l. 445 "intensity of /da/ would be at max 0.3. and of /ga/ 0.7".

7. Table 3: I feel that the first prediction is not actually a prediction, but a result of the article, as the first data shows that "The more predictable a word, the earlier this word is uttered."

8. Table 3 and Figure 8A: I think that the second prediction that "When there is a flat constraint distribution over an utterance (e.g., when probabilities are uniform over the utterance) the acoustics of speech should naturally be more rhythmic (Figure 8A)." could be tested with the current data. In the speech corpus, are sentences with lower linguistic constraints more rhythmic?

9. Table 3: "If speech timing matches the internal language model, brain responses should be more rhythmic even if the acoustics are not (Figure 8A)." What do the authors mean by "more rhythmic"? Does it mean the brain follows more accurately the acoustics? Does it generate stronger internal rhythms that are distinct from the acoustic temporal structure?

10. Figure 5 C: "Strength of 4 Hz power": what is the frequency bandwidth?

11. Figure 5 D: " Slice of D", Slice of C?

12. L 442: "propotions » -> proportions.

13. Abstract: "Our results reveal that speech tracking does not only rely on the input acoustics but instead entails an interaction between oscillations and constraints flowing from internal language model " I think this claim is too strong, considering that the article does not present direct electrophysiological evidence.

*Reviewer #2 (Recommendations for the authors):*

1. In the model the predictability is computed at the word-level, while the oscillator operates at the syllable level. The authors show different duration effects for syllables within words, likely related to predictability. Is there any consequence of this mismatch of scales?

2. Furthermore, could the authors clarify whether or not and how they think the model mechanism is different from top-down phase reset (e.g. l. 41). It seems that the excitability cycle at the intermediate word-level is shifted from being aligned to the 4 Hz oscillator though the linguistic feedback from layer l+1. Would that indicate a phase resetting at the word-level layer through the feedback?

3. The model shows how linguistic predictability can affect neuronal excitability in an oscillatory model, allowing to improve the processing of non-isochronous speech. I do not fully understand the claim that the linguistic predictability makes the processing (at the word-level) more isochronous, and why such isochronicity is crucial.

4. The authors showed that word frequency affects the duration of a word. Now the RNN model relates the predictability of a word (output) to the duration of the previous word W-1 (l. 187). Didn't one expect from Figure 1B that the duration of the actually predicted word is affected? How are these two effects related?

5. Title: is "constrained" the right word here, rather "modulated"? As we can process non-predictable speech.

6. See l. 129: "In this way, oscillations do not have to shift their phase after every speech unit and can remain at a relatively stable frequency as long as the internal model of the speaker matches the internal model of the perceiver." It seems to me that in the model the authors introduce, the phase-shifting still occurs. Even though the oscillator component is fixed, the activation threshold fluctuations at the word-level are "shifted" due to the feedback. So there is no feedforward phase-reset, however, a phase-reset due to feedback?

7. l. 219: why was bigrams added as control variable?

8. l. 233 in l. 142 it says that only 2848 words were present in CELEX. Where the 4837 sentences consisting of the 2848 words?

9. Figure 2 D,E the labeling with ρ and p is confusing, I'd at least state consistently both, so one sees the difference.

10. Table 1 legend: could you add why the specific transformations were performed?

11. l. 204: the β coefficient is rather small compared to the duration of W-1 effect. The dependent variable onset-to-onset should be strongly correlated with the W-1 duration. I wonder if this is a problem?

12. l. 249: what is meant with "after the first epoch"?

13. l. 254: how local were these lengthening effects? Did the predictability based on the trained RNN strongly vary across words or rather vary on a larger scale i.e. full sentences being less predictable than others?

14. l. 268: Could you explain where the constants are coming from: like the 20 and 100 ms windows for inhibition and the values -0.2 and -3. The function inhibit(ta) is not clear to me. What is the output when Ta is 0 versus 1?

15. Figure 4: the legend is very short, adding some description what the figure illustrates would make it easier to follow. The small differences in early/late activation are hard to see, particularly for the 4th row. Maybe it would help to add lines?

16. Figure 5 B: could you clarify the effect at late stim times relative to isochronous, i.e. why the supra time relative to isochronous decreases for highly predictable stimuli. I assume this is to the inhibition function?

17. How is the connectivity between layers defined? Is it symmetric for feedforward and feedback?

18. l. 294/l. 205: "with a delay of 0.9*ω seconds, which then decays at 0.01 unit per millisecond and influences the l-level at a proportion of 1.5." where are the constants coming from?

19. l. 347: "the processing itself can actually be closer to isochronous than what can be solely extracted from the stimulus". This refers to Figure 5 D I assume. Did you directly compare the acoustics and the model output with respect to isochrony?

20. l. 437-438: I am not fully understanding these choices: why is N1 represented by N2? Why is the probability of da and ga uneaven, and why are there nodes for da and ga (Nda, Nga) plus a node N2 which predicts both with different probability?

21. Figure 5: why is the power of the high-high predictable condition the lowest. Is this an artifact of the oscillator in the model being fixed at 4 Hz or related to the inhibition function? High-high should like low-low result in rather regular, but faster acoustics?

22. l. 600: "The perceived rhythmicity" In my view speech has been suggested to be quasi-rhythmic, as (1) some consistency in syllable duration has been observed within/across languages, and (2) as (quasi-)rhythmicity seemed a requirement to explain how segmentation of speech based on oscillations could work in the absence of simple segmentation cues (i.e. pauses between syllables). While one can ask when something is "rhythmic enough" to be called rhythmic, I don't understand why this is related to "perceived rhythmicity".

23. l. 604: interesting thought!

*Reviewer #3 (Recommendations for the authors):*

1. An important question is how the authors relate these findings to the Giraud and Poeppel, 2012 proposal which really focuses on the syllable. Would you alter the hypothesis to focus on the word level? Or remain at the syllable level and speed up and low down the oscillator depending on the predictability of each word? It would be interesting to hear the authors thoughts on how to manage the juxtaposition of syllable and word processing in this framework.

2. The authors describe the STiMCON model as having an oscillator with frequency set to the average stimulus rate of the sentence. But how an oscillator can achieve this on its own (without the hand of its overloads) is unclear particularly given a pseudo-rhythmic input. The authors freely accept this limitation. However, it is worth noting that the ability for an oscillator mechanism to do this under pseudorhythmic context is more complicated than it might seem, particularly once we include that the stimulus rate might change from the beginning to the end of a sentence and across an entire discourse.

3. The analysis of the naturalistic dataset shows a nice correlation between the estimated time shifts predicted by the model and the true naturalistic deviations. However, I find it surprising that there is so little deviation across the parameters of the oscillator (Figure 6A). What should we take from the fact that an oscillator aligned in anti-phase from the with the stimulus (which would presumably show the phase code only stimulus offsets), still shows a near equal correlation with true timing deviations. Furthermore, while the R2 shows that the predictions of the model co-vary with the true values, I'm curious to know how accurately they are predicted overall (in terms of mean squared error for example). Does the model account for deviations from rhythmicity of the right magnitude?

4. Lastly, it is unclear to what extent the oscillator is necessary to find this relative time shift. A model comparison between the predictions of the STiMCON and the RNN predictions on their own (à la Figure 3) would help to show how much the addition of the oscillation improves our predictions. Perhaps this is what is meant by the "non-transformed R2" but this is unclear.

5. Figure 7 shows a striking result demonstrating how the model can be used to explain an interesting finding that phase of an oscillation can bias perception towards da or ga. The initial papers consider this result to be explained by delays in onset between visual and auditory stimuli whereas this result explains it in terms of the statistical likelihood each syllable. It is a nice reframing which helps me to better understand the previous result.

6. The authors show that syllable lengths are determined in part by the predictability of the word it is a part of. While the authors have reasonably restricted themselves to a single hierarchical level, the point invites the question as to whether all hierarchical levels are governed by similar processes. Should syllables accelerate from beginning to end of a word? Or in more or less predictable phrases?

7. Figure 5 shows how an oscillator mechanism can force pseudo-rhythmic stimuli into a more rhythmic code. The authors note that this can be done either by slowing responses to early stimuli and quickening responses to later ones, or by dropping (nodes don't reach threshold) stimuli too far outside the range of the oscillation. The first is an interesting mechanism, the second is potentially detrimental to processing (although it could be used as a means for filtering out noise). The authors should make clear how much deviation is required to invoke the dropping out mechanism and how this threshold relates to the naturalistic case. This would give the reader a clearer view of the flexibility of this model.

8. I found Figure 5 very difficult to understand and had to read and read it multiple times to feel like I could get a handle on it. I struggled to get a handle on why supra time was shorter and shorter the later the stimulus was activated. It should reverse at some point as the phase goes back into lower excitability, right? The current wording is very unclear on this point. In addition, the low-high, high-low analysis is unclear because the nature of the stimuli is unclear. I think an added figure panel to show how these stimuli are generated and manipulated would go a long way here.

9. The prediction of behavioral data in Figure 7 is striking but the methods could be improved. Currently, the authors bin the output of the model to be 0, 0.5 or 1 which requires some maneuvering to effectively compare it with the sinewave model. They could instead use a continuous measure (either lag of activation between da and ga, or activation difference) as a feature in a logistic regression to predict the human subject behavior.

10. I'm not sure but I think there is a typo in line 383-384. The parameter for feedback should read Cl+1◊ l * Al+1,T. Note the + sign instead of the -. Or I have misunderstood something important.

[Editors' note: further revisions were suggested prior to acceptance, as described below.]

Thank you for resubmitting your work entitled "Tracking of pseudo-rhythmic speech is modulated by linguistic predictions in an oscillating computational model" for further consideration by *eLife*. Your revised article has been evaluated by Andrew King (Senior Editor) and a Reviewing Editor.

The manuscript has been greatly improved, and only these issues need to be addressed, as outlined below:

*Reviewer #2 (Recommendations for the authors):*

I want to thank the authors for the great effort revising the manuscript. The manuscript has much improved. I only have some final small comments.

Detailed comments

l. 273-275: In my opinion: This is because the oscillator is set as a rigid oscillator in the model that is not affected by the word level layer activation; however, as the authors already discuss this topic, this is just a comment.

l. 344: "the processing itself" I'd specify: "the processing at the word layer".

l. 557/558: Rimmele et al., (2018) do discuss that besides the motor system, predictions from higher-level linguistic processing might affect auditory cortex neuronal oscillations through phase resetting. Top-down predictions affecting auditory cortex oscillations is one of the main claims of the paper. Thus, this paper seems not a good example for proposals that exclude when-to-what interactions. In my view the claims are rather consistent with the ones proposed here, although Rimmele et al., do not detail the mechanism and differ from the current proposal in that they suggest phase resetting. Could you clarify?

l 584 ff.: "This idea diverges from the idea that entrainment should per definition occur on the most excitable phase of the oscillation [3,15]." Maybe rephrase: "This idea diverges from the idea that entrainment should align the most excitable phase of the oscillation with the highest energy in the acoustics [3,15]."

l. 431: "The model consists of four nodes (N1, N2, Nda, and Nga) at which N1 activation predicts a second unspecific stimulus (S2) represented by N2 at a predictability of 1. N2 activation predicts either da or ga at 0.2 and 0.1 probability respectively."

This is still hard to understand for me. E.g. What is S2, is this either da or ga, wouldn't their probability have to add up to 1?

Wording

l. 175/176: sth is wrong with the sentence.

l. 544: "higher and syllabic"? (sounds like sth is wrong in the wording)

l. 546: "within more frequency" (more frequent or higher frequency?)

---

## [Author Response]

Essential revisions:All reviewers had a very positive assessment of the manuscript. They find the described work highly novel and interesting. However, they also find the manuscript quite dense and complex, and suggest some clarifications in the description of the model and in the methods.Please find a list of the recommendations below.Reviewer #1 (Recommendations for the authors):1. First of all, I think that the concept of "internal language model" should be defined and described in more detail in the introduction. What does it mean when it is weak and when it is strong, for the producer and for the receiver?

We define internal language model as the individually acquired statistical and structural knowledge of language stored in the brain. A virtue of such an internal language model is that it can predict the most likely future input based on the currently presented speech information. If a language model creates strong predictions, we call it a strong model. In contrast, a weak model creates no or little predictions about future input (note that the strength of individual predictions depends not only on the capability of the system to create a prediction, but also on the available information). If a node represents a speech unit that is likely to be spoken next, a strong internal language model will sensitize this node and it will therefore be active earlier, that is, on a less excitable phase of the oscillation.

The above explanation has been included in the introduction.

2. It is still not fully clear to me what kind of information the internal oscillation is entraining to in the model. From Figure 1, it seems that the oscillation is driven by the acoustics only, but the phase of processing of linguistic units depends on their predictability.

The entrainment proper is indeed still the acoustics. However, compared to other models in which oscillations are very strongly coupled to this acoustic envelope by aligning the most excitable phase as a consequence of the acoustic phase shifts (Doelling et al., 2019) or proactively dependent on temporal predictions (Rimmele et al., 2018), we propose that it is not necessary to change the phase of the ongoing oscillation in response to the acoustics to optimally process pseudo-rhythmic speech. As such, we view the model as weakly coupled relatively to more strongly coupled oscillator models. The phase of the oscillation does not need to change to every phase shift in the acoustics. We propose that the model can entrain to the average speech rate, however, we acknowledge in the updated manuscript we do not answer how this can be done. We have added a discussion on this point in the discussion which now reads: “We aimed to show that a stable oscillator can be sensitive to temporal pseudo-rhythmicities when these shifts match predictions from an internal linguistic model (causing higher sensitivity to these nodes). In this way we show that temporal dynamics in speech and the brain cannot be isolated from processing the content of speech. This is contrast with other models that try to explain how the brain deals with pseudo-rhythmicity in speech (Giraud and Poeppel, 2012, Rimmele et al., 2018, Doelling et al., 2019), which typically do not take into account that the content of a word can influence the timing. Phase resetting models can only deal with pseudo-rhythmicity by shifting the phase of ongoing oscillations in response to a word that is off-set to the mean frequency of the input (Giraud and Poeppel, 2012, Doelling et al., 2019). We believe that this goes beyond the temporal and content information the brain can extract from the environment which has what/when interactions. However, our current model does not have an explanation of how the brain can actually entrain to an average speech rate. This is much better described in dynamical systems theories in which this is a consequence of the coupling strength between internal oscillations and speech acoustics (Doelling et al., 2019, Assaneo et al., 2021). However, these models do not take top-down predictive processing into account. Therefore, the best way forward is likely to extend coupling between brain oscillations and speech acoustics (Poeppel and Assaneo, 2020) with the coupling of brain oscillations to brain activity patterns of internal models (Cumin and Unsworth, 2007).”

3. If acoustic information arrives at the most excitable phase of the neural oscillation (as described in figure 1), and if predictability makes words arrive earlier, does it entail that more predictable words arrive at less excitable phases of the neural oscillation? What would be the computational advantage of this mechanism?

Indeed, more predictable words arrive at a less excitable phase of the oscillation. This is an automatic consequence of the statistics in the environment in which more predictable words are uttered earlier. The brain could utilize these statistical patterns (we can infer that earlier uttered words are more predictable). If we indeed do this, temporal information in the brain contains content information. But how would the brain code for this? We propose that this can be done by phase-of-firing, such that the phase at which neurons are active is relevant for the content of the item to be processed. Computationally it would be advantageous to be able to separate different representations in the brain by virtue of phase coding. As such, you don’t only have a spatial code of information (which neuron is active), but also a temporal code of information (when is a neuron active). This leads to a redundancy of coding which has a lot of computational advantages in a noisy world (e.g. Barlow, 2001).

The idea of phase coding has been proposed in the past (Jensen and Lisman, 2005; Kayser et al., 2009; Panzeri et al., 2001 Hopfield, 1995). Indeed, there is evidence that time or phase of firing contains content information (Siegel et al., 2012; O’Keefe and Recce, 1993). In some of these theoretical accounts, the first active node actually provides more information that the less-specific activation occurring later (Hopfield, 1995). One might say a neuron that is active already at a low excitable phase contains more information about the relevance of the activation than many neurons that are going to be active at a very excitable phase. Indeed, neurons in rest are found to be active at excitable phases (Haegens et al., 2011). In a similar vein, it has also been suggested that α power/phase modulates not the sensitivity, but mostly the bias to detect something (Iemi et al., 2016), suggesting that high excitable points do not improve sensitivity, but merely the idea that something was perceived. In sum, computational advantages of this model relate to having information about time relate to information on content by virtue of a phase code. To make this clear we have added a section in the discussion. It now reads: “A natural feature of our model is that time can act as a cue for content implemented as a phase code (Jensen et al., 2012, Lisman and Jensen, 2013). This code unravels as an ordered list of predictability strength of the internal model. This idea diverges from the idea that entrainment should per definition occur on the most excitable phase of the oscillation (Giraud and Poeppel, 2012, Rimmele et al., 2018). Instead, this type of phase could increase the brain representational space to separate information content (Panzeri, Petersen et al. 2001, Lisman and Jensen, 2013). We predict that if speech nodes have a different base activity, ambiguous stimulus interpretation should be dependent on the time/phase of presentation (see (Ten Oever and Sack, 2015, Ten Oever et al., 2020))”.

4. What is "stimulus intensity" in figure 5? Does it reflect volume or SNR?

We have added this information in the figure legend. It reflects the overall amplitude of the input in the model. See Figure 5A. We now consistently refer to intensity when referring to the amplitude of the input.

5. Similarly what is "amplitude" in Figure 6?

Amplitude refers to the amplitude of the sinus in the model. This information is now in the figure legend.

6. L. 376-L. 439 "N2 activation predicts either da or ga at 0.2 and 0.1 probability respectively." Please explain why the probabilities are not equal in the model, same for l. 445 "intensity of /da/ would be at max 0.3. and of /ga/ 0.7".

We regret that this was unclear. Phase coding of information only occurs when the internal model of STiMCON has different probabilities of predicting the content of the next word. Otherwise, the nodes will be active at the same time. The assumption for the /da/ and /ga/ having different probabilities is reasonable as the /d/ and /g/ consonant have a different overall proportion in the Dutch language (with /d/ being more frequent than the /g/). As such, we would expect the overall /d/ representation in the brain to be active at lower thresholds than then the /g/ representation. We have clarified this now in the manuscript: “N2 activation predicts either da or ga at 0.2 and 0.1 probability respectively. This uneven prediction of /da/ and /ga/ is justified as /da/ is more prevalent in the Dutch language as /ga/ (Zuidema, 2010) and it thus has a higher predicted level of occurring.”

l. 445 refers to what input we gave to the model. We gave the model the input for a fully unambiguous /da/, a fully unambiguous /ga/, and morphs in between. This was to demonstrate the behavior of the model to show that only at the ambiguous stimulation we would find a phase code of information.

7. Table 3: I feel that the first prediction is not actually a prediction, but a result of the article, as the first data shows that "The more predictable a word, the earlier this word is uttered."

We agree and have removed it from the table.

8. Table 3 and Figure 8A: I think that the second prediction that "When there is a flat constraint distribution over an utterance (e.g., when probabilities are uniform over the utterance) the acoustics of speech should naturally be more rhythmic (Figure 8A)." could be tested with the current data. In the speech corpus, are sentences with lower linguistic constraints more rhythmic?

We thank the reviewer for the interesting suggestion. We indeed do have the data to investigate whether acoustics are more rhythmic when they are less predictable across the sentence. To investigate this question, we extracted the RNN prediction for 10 subsequent words. Then we extracted the variance of the prediction across those 10 words and extracted the word onset itself. We created a time course at which word onset were set to 1 (at a sampling rate of 100 Hz). Then we performed an FFT and extracted z-transformed power values over a 0-15 Hz interval. The power at the maximum power value with the theta range (3-8Hz) was extracted. These max z-scores were correlated with the log transform of the variance (to normalize the skewed variance distribution; Figure 3E). We found a weak, but significant negative correlation (r = -0.062, p < 0.001; Figure 3D) in line with our hypothesis. This suggests that the more variable the predictions within a sentence, the lower the peak power value is. When we repeated the analysis on the envelope, we did not find a significant effect. We have added this analysis in the main manuscript and added two panels to figure 2.

We do believe that a full answer to this question requires more experimental work and therefore keep it as a prediction in the table.

9. Table 3: "If speech timing matches the internal language model, brain responses should be more rhythmic even if the acoustics are not (Figure 8A)." What do the authors mean by "more rhythmic"? Does it mean the brain follows more accurately the acoustics? Does it generate stronger internal rhythms that are distinct from the acoustic temporal structure?

This would refer to the brain having a stronger isochronous response for non-isochronous than isochronous acoustics. This is based on the results in figure 6 (previous figure 5). In figure 6 we show that when the internal model predicts the next word at alternatingly high or low predictabilities, the model’s response is not most isochronous (strongest 4 Hz response) when the acoustics are isochronous, but rather when the acoustics are shifted in line with the internal model (more predictable words occurring earlier). We would expect the same in the brain’s responses. We think the wording rhythmic is not correct in this context and should rather refer to isochronous. We have updated the text to now refer to isochrony.

10. Figure 5 C: "Strength of 4 Hz power": what is the frequency bandwidth?

This is the peak activity. We clarified this now in the text. You can see from (current) Figure 6A+D that the response overall is also very peaky (by nature of the stimulation and the isochrony of the oscillation that we enter in the model).

11. Figure 5 D: " Slice of D", Slice of C?

This is the peak activity. We clarified this now in the text. You can see from (current) Figure 6A+D that the response overall is also very peaky (by nature of the stimulation and the isochrony of the oscillation that we enter in the model).

12. L 442: "propotions » -> proportions.

Adjusted accordingly.

13. Abstract: "Our results reveal that speech tracking does not only rely on the input acoustics but instead entails an interaction between oscillations and constraints flowing from internal language model " I think this claim is too strong, considering that the article does not present direct electrophysiological evidence.

We agree and regret this strong claim. We have updated the text and it now reads: “Our results suggest that speech tracking does not have to rely only on the acoustics but could also entail an interaction between oscillations and constraints flowing from internal language models.”

Reviewer #2 (Recommendations for the authors):1. In the model the predictability is computed at the word-level, while the oscillator operates at the syllable level. The authors show different duration effects for syllables within words, likely related to predictability. Is there any consequence of this mismatch of scales?

The current model does indeed operate on the word level, while oscillatory models operate on the syllabic level. We do not claim by this that predictions per see only work on a word level. In contrary, we believe that ultimately also syllabic level predictions as well as higher level linguistic predictions can be made to influence speech processing. Therefore, our model is incomplete, but serves the purpose to demonstrate how internal language models can influence speech timing as well as perceptual tracking.

Our choice of the word level was mostly practical. We choose in the current manuscript to start with a word level prediction as this is the starting point commonly available and applied for RNNs. RNNs often work on the word level and not on the syllabic level. For example, this allowed use to use highly trained word level embeddings as a starting point for our LSTM. We are not aware of pre-trained syllabic embeddings that can achieve the same thing. As mentioned above, the temporal shift in STiMCON would also be predicted based on syllabic prediction. Therefore, the only results that are really affected by this notion are the results of figure 6 (or current figure 7). Predicting the temporal shift would likely have been benefitted from also adding predictions in temporal shifts based on syllabic and higher order linguistic predictions.

We now add a note about the level of processing that might affect the results of current figure 7 as well as a paragraph in the discussion that mentions that the model should have multiple levels operating at different levels of the linguistic hierarchy.

The result section now reads: “Note that the current prediction only operated on the word node (to which we have the RNN predictions), while full temporal shifts are probably better explained by word, syllabic and phrasal predictions.”

The discussion now reads: “The current model is not exhaustive and does not provide a complete explanation of all the details of speech processing in the brain. For example, it is likely that the primary auditory cortex is still mostly modulated by the acoustic pseudo-rhythmic input and only later brain areas follow more closely the constraints posed by the language model of the brain. Moreover, we now focus on the word level, while many tracking studies have shown the importance of syllabic temporal structure (Luo and Poeppel, 2007, Ghitza, 2012, Giraud and Poeppel, 2012) as well as the role of higher order linguistic temporal dynamics (Meyer, Sun et al. 2019, Kaufeld et al., 2020).”

2. Furthermore, could the authors clarify whether or not and how they think the model mechanism is different from top-down phase reset (e.g. l. 41). It seems that the excitability cycle at the intermediate word-level is shifted from being aligned to the 4 Hz oscillator though the linguistic feedback from layer l+1. Would that indicate a phase resetting at the word-level layer through the feedback?

The model is different as it does not assume that the top-down influence needs to reflect a phase reset. A phase reset would indicate a shift in the ongoing phase of the oscillator (Winfree 2001). However, the feedback in our model does not shift the phase of the oscillator. The phase of the oscillator remains stable, only the phase at which a node in a model is active changes. The feedback is implemented as a constant input that decays over time, it does not actively interfere with the phase of the oscillator at the word level. We propose here that if the internal model of the perceiver and receiver perfectly align the oscillator can remain entrained to the average speaker rate without shifting phase at every word, but instead can take the linguistic predictions into account.

We don’t claim phase resetting does not occur. We have acknowledged that our model does not account for this (yet) in the discussion. However, in our model phase resetting would be required when the predictions of perceiver and receiver don’t match through which the expected timing don’t match. To clarify the difference with phase reset models we have added a section in the discussion. It now reads: “We aimed to show that a stable oscillator can be sensitive to temporal pseudo-rhythmicities when these shifts match predictions from an internal linguistic model (causing higher sensitivity to these nodes). In this way we show that temporal dynamics in speech and the brain cannot be isolated from processing the content of speech. This is contrast with other models that try to explain how the brain deals with pseudo-rhythmicity in speech (Giraud and Poeppel 2012, Rimmele et al., 2018, Doelling et al., 2019), which typically do not take into account that the content of a word can influence the timing. Phase resetting models can only deal with pseudo-rhythmicity by shifting the phase of ongoing oscillations in response to a word that is off-set to the mean frequency of the input (Giraud and Poeppel, 2012, Doelling et al., 2019). We believe that this goes beyond the temporal and content information the brain can extract from the environment which has what/when interactions. However, our current model does not have an explanation of how the brain can actually entrain to an average speech rate. This is much better described in dynamical systems theories in which this is a consequence of the coupling strength between internal oscillations and speech acoustics (Doelling et al., 2019, Assaneo et al., 2021). However, these models do not take top-down predictive processing into account. Therefore, the best way forward is likely to extend coupling between brain oscillations and speech acoustics (Poeppel and Assaneo, 2020) with the coupling of brain oscillations to brain activity patterns of internal models (Cumin and Unsworth, 2007).”

3. The model shows how linguistic predictability can affect neuronal excitability in an oscillatory model, allowing to improve the processing of non-isochronous speech. I do not fully understand the claim that the linguistic predictability makes the processing (at the word-level) more isochronous, and why such isochronicity is crucial.

The main point of figure 5 (current figure 6) is to show that acoustic time and brain time do not necessarily have to have the same relation. We show that isochronously presented acoustic input does not need to lead to the most isochronous brain responses. Why does the model behave this way? Effectively the linguistic feedback changes the time at which different word nodes are active and thereby does not make it directly linked to the acoustic input. This creates an interesting dynamic that causes the model’s response to follow a stronger 4 Hz response for non-isochronous acoustic input. This dynamic was not necessarily predicted by us in advance, but is what came out of the model. The exact dynamics can be seen in Figure 6 —figure supplement 2. It seems that due to the interactions between acoustic input and feedback, higher peak activation at a 4 Hz rhythm is reached at an acoustic offset that matches the linguistic predictions (earlier presentation when words are more predictable).

We don’t necessarily believe isochronicity is crucial for the brain to operate and it can deviate from isochronicity. But isochronicity provides two clear benefits: (1) strong temporal predictions and (2) higher processing efficiency (Schroeder and Lakatos, 2009). If it is possible for the brain to maintain a stable oscillator, this would increase the efficiency relative to changing its phase after the presentation of every word. If the brain needs to shift its phase after every word, we even wonder what the computational benefit of an oscillator is. If simply the acoustics are followed, why would we need an oscillator at all?

We claim that the brain shows a more isochronous response when the linguistic predictions match the temporal shifts (earlier onsets for more predictable words). If this is not the case the isochrony is also lower. As said above, this provides the benefit that there is higher processing efficiency. But in reality, the predictions of a receiver will not always match the exact timing of the production of the producer. Therefore, some phase shifts are likely still needed (not modelled here).

4. The authors showed that word frequency affects the duration of a word. Now the RNN model relates the predictability of a word (output) to the duration of the previous word W-1 (l. 187). Didn't one expect from Figure 1B that the duration of the actually predicted word is affected? How are these two effects related?

We expect that when words are easier accessible, they are uttered early. This is why we created the RNN based on prediction the onset of the next word (see prediction in Figure 1B). However, based on previous literature in deed it could also be expected that word duration itself is affected by the predictability (Lehiste, 1972, Pluymaekers et al., 2005). However, most of these effects have been shown on mean duration of words and not that the duration of words within a sentence is modulated in the context of the full sentence above what can be explained by word frequency. Indeed, we replicate the findings of individual words showing that word frequency (likely affecting accessibility to a word) affect the mean duration of this word. To test if this extends within the context of predictability within a sentence, we reran our linear regression model, but used word duration as our dependent variable. As other control variables we included the bigram, the frequency of the current word, the mean duration of the current word and the syllable rate. Note that only effects will be significant in which the effects are stronger than can be expected based on the mean duration. Results can be found in Table 1.

We found no effect of the RNN prediction on the overall duration of word (above what can be explained by the other control factors). Other factors did show effects, such as the word frequency and bigram (p < 0.001). Interesting and unexpected, these resulted in longer duration, not shorter durations (positive t-values). At this moment we do not have an explanation for this effect, but it could be that this lengthening is a consequence of the earlier onset of the individual word by which a speaker tries to keep to the average rate. Alternatively, it is possible that this positive relation is a consequence of the predictability of the word following, such that words get shorter if the word after it is more predictable (as we showed in the manuscript). However, in this case we would also expect the RNN prediction to be significant. We have now reported on these effects in the supplementary materials, but do not have a direct explanation for the direction of this effect.

5. Title: is "constrained" the right word here, rather "modulated"? As we can process non-predictable speech.

This is fair and we changed it to modulated. Indeed, we can perfectly well process non-predictable speech as well.

6. See l. 129: "In this way, oscillations do not have to shift their phase after every speech unit and can remain at a relatively stable frequency as long as the internal model of the speaker matches the internal model of the perceiver." It seems to me that in the model the authors introduce, the phase-shifting still occurs. Even though the oscillator component is fixed, the activation threshold fluctuations at the word-level are "shifted" due to the feedback. So there is no feedforward phase-reset, however, a phase-reset due to feedback?

This is fair and we changed it to modulated. Indeed, we can perfectly well process non-predictable speech as well.

7. l. 219: why was bigrams added as control variable?

We regret this was unclear. We were interested to investigate if a bigger context (as measured with the RNN) provides more information than a bigram which only investigates the statistics regarding two previous words.

8. l. 233 in l. 142 it says that only 2848 words were present in CELEX. Where the 4837 sentences consisting of the 2848 words?

We did two rounds of checking our dictionary of words. In the first round we investigated if the words were present in the word2vec embeddings (otherwise we couldn’t use them for the RNN). If not, they were marked with a <unknown> label. The RNN was ran with all these words. This refers to the 4837 sentences. In total there were 3096 unique words.

For the regression analyses, we further wanted to extract parameters of the individual words. Thus, we investigated if the words were present in CELEX. This was the case of 2848 of 3096 words. For the regression analyses we only included the words so that we could estimate the relevant parameters. This was when the W-1 word was in CELEX (for the analysis in 4 this was related to the current word). For the regression we therefore didn’t include all the sentences going into the RNN.

9. Figure 2 D,E the labeling with ρ and p is confusing, I'd at least state consistently both, so one sees the difference.

We now also report on the p-value in the figure.

10. Table 1 legend: could you add why the specific transformations were performed?

The transformations were performed to ensure that our factors were close-to-normal in order to enter in the regression analyses. This information is now in the main manuscript. We would like to note that our analysis is robust against changes in the transformations. If we don’t perform any transformation the same results hold.

11. l. 204: the β coefficient is rather small compared to the duration of W-1 effect. The dependent variable onset-to-onset should be strongly correlated with the W-1 duration. I wonder if this is a problem?

Indeed, the word duration has the strongest relation to the onset-to-onset difference (as is of course intuitive, but also evident from the β coefficient). To capture this variance, we added this variable into the regression analyses. When performing a regression analyses it is useful to include factors that explain variance in your model. This ensure that the factor of interest (here RNN prediction) captures any variance that cannot be related to variance already explained by the other factors. Therefore, we don’t feel that this is a big problem, but actually an intended and expected effect.

12. l. 249: what is meant with "after the first epoch"?

We have clarified it. RNN are normally trained with different steps (or epochs). After every epoch the weights in the model are adjusted to reduce the overall error of the fit. But this term might be specific to keras, and not to machine learning in general. We now state: “entering the RNN predictions after the first training cycle (of a total of 100)”.

13. l. 254: how local were these lengthening effects? Did the predictability based on the trained RNN strongly vary across words or rather vary on a larger scale i.e. full sentences being less predictable than others?

To answer this question, we investigated for individual words and sentence positions whether the RNN prediction was generally higher. Indeed, for some words average prediction was higher. Both for the word that was predicted as well as for the last word before the prediction. For sentence position it seemed that for words very early in the sentence there was a lower overall prediction than for words later in the sentence. We have added a figure as a supporting figure to the manuscript.

14. l. 268: Could you explain where the constants are coming from: like the 20 and 100 ms windows for inhibition and the values -0.2 and -3. The function inhibit(ta) is not clear to me. What is the output when Ta is 0 versus 1?

The inhibition function reflects the excitation and inhibition of individual nodes in the model and is always relative to the time of activation of that node (Ta). When Ta is less than 20 (in ms), the node is activated (-3* inhibition factor), when Ta is between 20 and 100 there is strong inhibition, and after that, there is a base inhibition on the node. So, for Ta 0 versus 1 the output is both -3*BaseInhibition. The point of this activation function is to have a nonlinear activation, loosely resembling the nonlinear activation pattern in the brain.

The values for inhibition reflect rather early excitation (20 ms) and longer lasting inhibition (100 ms). We acknowledge that these numbers are only loosely related to neurophysiological time scales and is of course highly dependent on the region of interest and the local and distant connections. However, the exact timing is not critical to the main output factors of the model (phase coding and temporal sensitivity for higher predictable scales) as long as there is excitation and inhibition. We have added the rationale behind our parameter choice in the manuscript.

15. Figure 4: the legend is very short, adding some description what the figure illustrates would make it easier to follow. The small differences in early/late activation are hard to see, particularly for the 4th row. Maybe it would help to add lines?

We have clarified the description and have added a zoom-in on the relevant early/late activations for the critical word including relevant lines. We hope that this has improved the readability of this figure.

16. Figure 5 B: could you clarify the effect at late stim times relative to isochronous, i.e. why the supra time relative to isochronous decreases for highly predictable stimuli. I assume this is to the inhibition function?

This is indeed related to the inhibition function. As soon as high enough activation reaches a node, the node will reach suprathreshold activation due to the feedback (when Ta < 20). While the node is at suprathreshold activation, very low sensory input will push the node to reach suprathreshold as a consequence due to sensory input. See Author response image 1.

**Author response image 1. sa2fig1:** 

Here the ‘nice’ node is already activated due to the feedback (supra threshold activation panel is orange), then the sensory input arrives, and the node immediately reaches a 2 activation (red color. Activation due to sensory input). This of course never happens for stimuli that have no feedback (the ‘I’ node) and happens later for nodes that have weaker feedback (later for ‘very’ compared to ‘cake’). Moreover, after the supra threshold due to feedback is finished, the inhibition sets in reducing the activation and the nodes never (or at least at intensities used for the simulation) reaches threshold for later stimulus times during inhibition.

17. How is the connectivity between layers defined? Is it symmetric for feedforward and feedback?

In the current model the feedforward layers only connect to nodes representation the same items (i.e. a ‘I’ L-1 (stimulus level) node connects to a ‘I’ L node which connects to a ‘I’ L+1 node). Only the feedback nodes (L+1) are fully connected with the active level (L) but with different connection strengths which are defined by the internal model (defined in table 2).

18. l. 294/l. 205: "with a delay of 0.9*ω seconds, which then decays at 0.01 unit per millisecond and influences the l-level at a proportion of 1.5." where are the constants coming from?

These are informed choices. The 0.9*ω was defined as we hypothesized that onset time would be loosely predicted around on oscillatory cycle, but to be prepared for input slightly earlier (which of course happens for predictable stimuli), we set it to 0.9 times the length of the cycle. The decay is needed and set such that the feedback would continue around a full theta cycle. The proportion was set empirically such to ensure that strong feedback did cause suprathreshold activation at the active node. We added this explanation to the manuscript.

19. l. 347: "the processing itself can actually be closer to isochronous than what can be solely extracted from the stimulus". This refers to Figure 5 D I assume. Did you directly compare the acoustics and the model output with respect to isochrony?

We can compare the relative distribution (so when is the peak strongest across delays), but not the absolute values as the stimulus intensity and the activation are not on the same unit scale. In order to promote this comparison, we now also show the power of the stimulus input distribution across stimulus intensities and delays (Figure 6B). It is evident that the stimulus has a symmetrical 4 Hz power spectral which is strongest at a delay of 0 (isochrony). This is not strange as we defined it a-priory this way.

20. l. 437-438: I am not fully understanding these choices: why is N1 represented by N2? Why is the probability of da and ga uneaven, and why are there nodes for da and ga (Nda, Nga) plus a node N2 which predicts both with different probability?

N1 represent N1. We regret the confusing, we meant to state that N1 predicts N2. N2 is represented as N2. We have rephrased this sentence to clarify it.

The probabilities of /da/ and /ga/ are uneven as the model only shows phase coding of information when the internal model of STiMCON has different probabilities of predicting the content of the next word. Otherwise, the nodes will be active at the same time. The assumption for the /da/ and /ga/ having different probabilities is reasonable as the /d/ and /g/ consonant have a different overall proportion in the Dutch language (with /d/ being more frequent than the /g/). As such, we would expect the overall /d/ representation in the brain to be active at lower thresholds than then the /g/ representation. We have clarified this now in the manuscript. It now reads: “N2 activation predicts either da or ga at 0.2 and 0.1 probability respectively. This uneven prediction of /da/ and /ga/ is justified as /da/ is more prevalent in the Dutch language as /ga/ [64] and it thus has a higher predicted level of occurring.”

21. Figure 5: why is the power of the high-high predictable condition the lowest. Is this an artifact of the oscillator in the model being fixed at 4 Hz or related to the inhibition function? High-high should like low-low result in rather regular, but faster acoustics?

We thank the reviewer and regret an unfortunate mistake. The labeling of the low-low and high-high was reversed. So actually, the high-high has the stronger activation. We have updated the labels and text accordingly.

Regarding the interpretation of the reviewer, we do indeed predict in natural situations for high-high that the acoustics to be slightly faster. However, in the simulations in figure 5 (current figure 6), the speed of the acoustics is never modulated, but decided by us in the simulation. Therefore, only the response of the model is estimated to varying internal models is estimated.

22. l. 600: "The perceived rhythmicity" In my view speech has been suggested to be quasi-rhythmic, as (1) some consistency in syllable duration has been observed within/across languages, and (2) as (quasi-)rhythmicity seemed a requirement to explain how segmentation of speech based on oscillations could work in the absence of simple segmentation cues (i.e. pauses between syllables). While one can ask when something is "rhythmic enough" to be called rhythmic, I don't understand why this is related to "perceived rhythmicity".

We regret our terminology. We mean perceived isochronicity. Indeed, rhythmicity occurs without isochrony (Obleser, Henry and Lakatos, 2017) and we don’t intend to state that natural speech timing do not have rhythm and pushing away from isochrony does not reflect rhythm. We mere meant to state that when the brain response is more isochronous (relative to acoustic stimulation), than likely the perception is also more isochronous.

23. l. 604: interesting thought!

We hope to pursue these ideas in the future.

Reviewer #3 (Recommendations for the authors):1. An important question is how the authors relate these findings to the Giraud and Poeppel, 2012 proposal which really focuses on the syllable. Would you alter the hypothesis to focus on the word level? Or remain at the syllable level and speed up and low down the oscillator depending on the predictability of each word? It would be interesting to hear the authors thoughts on how to manage the juxtaposition of syllable and word processing in this framework.

The current model does indeed operate on the word level, while oscillatory models operate on the syllabic level. We do not claim by this that predictions per see only work on a word level. In contrary, we believe that ultimately also syllabic level predictions as well as higher level linguistic predictions can be made to influence speech processing. Therefore, our model is incomplete, but serves the purpose to demonstrate how internal language models can influence speech timing as well as perceptual tracking.

Our choice of the word level was mostly practical. We choose in the current manuscript to start with a word level prediction as this is the starting point commonly available and applied for RNNs. RNNs often work on the word level and not on the syllabic level. For example, this allowed use to use highly trained word level embeddings as a starting point for our LSTM. We are not aware of pre-trained syllabic embeddings that can achieve the same thing. As mentioned above, the temporal shift in STiMCON would also be predicted based on syllabic prediction. Therefore, the only results that are really affected by this notion are the results of figure 6 (or current figure 7). Predicting the temporal shift would likely have been benefitted from also adding predictions in temporal shifts based on syllabic and higher order linguistic predictions.

We would predict that linguistic inference involves integrating knowledge at different hierarchical levels including predictions concerning which syllable comes next, based on syllabic regularities, word-to-word regularities as well as syntactic context. This is definitely on our to-do list to extrapolate the model to include predictions at these different layers. To ensure this is clear to the reader, we now add a note about the level of processing that might affect the results of figure 7 as well as a paragraph in the discussion that mentions that the model should have multiple levels operating at different levels of the linguistic hierarchy.

The result section now reads: “Note that the current prediction only operated on the word node (to which we have the RNN predictions), while full temporal shifts are probably better explained by word, syllabic and phrasal predictions.”

The discussion now reads: “The current model is not exhaustive and does not provide a complete explanation of all the details of speech processing in the brain. For example, it is likely that the primary auditory cortex is still mostly modulated by the acoustic pseudo-rhythmic input and only later brain areas follow more closely the constraints posed by the language model of the brain. Moreover, we now focus on the word level, while many tracking studies have shown the importance of syllabic temporal structure (Luo and Poeppel, 2007, Ghitza, 2012, Giraud and Poeppel, 2012) as well as the role of higher order linguistic temporal dynamics (Meyer et al., 2019, Kaufeld et al., 2020).”

2. The authors describe the STiMCON model as having an oscillator with frequency set to the average stimulus rate of the sentence. But how an oscillator can achieve this on its own (without the hand of its overloads) is unclear particularly given a pseudo-rhythmic input. The authors freely accept this limitation. However, it is worth noting that the ability for an oscillator mechanism to do this under pseudorhythmic context is more complicated than it might seem, particularly once we include that the stimulus rate might change from the beginning to the end of a sentence and across an entire discourse.

This is a clear limitation, but to be fair also a limitation of any proposal on entrainment not often addressed (Giraud and Poeppel, 2012; Ghitza, 2012). The reviewer himself has done great work on investigating oscillatory entrainment based on ideas on dynamical systems and weakly coupled oscillators (Doelling et al., 2019). These simulations show that oscillations can shift their frequency when the coupling to the acoustics is strong enough. If the coupling is very strong, this could even happen from word to word (as the intrinsic oscillator in that case almost one-to-one follows the phase of the acoustics). However, our contribution here is that none of these models take top-down influences about word predictability into account on these dynamics. Moreover, it is difficult to explain how these models can explain how temporal information can inform about the content of a word (Ten Oever et al., 2013; Ten Oever and Sack, 2015). Specifically, the effect on how syllable identity depends on oscillatory phase (Ten Oever and Sack, 2015). We added a section on this in the discussion clarifying this limitation and our contribution relating to this type of research.

It now reads: “We aimed to show that a stable oscillator can be sensitive to temporal pseudo-rhythmicities when these shifts match predictions from an internal linguistic model (causing higher sensitivity to these nodes). In this way we show that temporal dynamics in speech and the brain cannot be isolated from processing the content of speech. This is contrast with other models that try to explain how the brain deals with pseudo-rhythmicity in speech (Giraud and Poeppel, 2012, Rimmele et al., 2018, Doelling et al., 2019), which typically do not take into account that the content of a word can influence the timing. Phase resetting models can only deal with pseudo-rhythmicity by shifting the phase of ongoing oscillations in response to a word that is off-set to the mean frequency of the input (Giraud and Poeppel, 2012, Doelling et al., 2019). We believe that this goes beyond the temporal and content information the brain can extract from the environment which has what/when interactions. However, our current model does not have an explanation of how the brain can actually entrain to an average speech rate. This is much better coupling in dynamical systems theories in which this is a consequence of the coupling strength between internal oscillations and speech acoustics (Doelling et al., 2019, Assaneo et al., 2021). However, these models do not take top-down predictive processing into account. Therefore, the best way forward is likely to extend coupling between brain oscillations and speech acoustics (Poeppel and Assaneo, 2020) with the coupling of brain oscillations to brain activity patterns of internal models (Cumin and Unsworth, 2007).”

3. The analysis of the naturalistic dataset shows a nice correlation between the estimated time shifts predicted by the model and the true naturalistic deviations. However, I find it surprising that there is so little deviation across the parameters of the oscillator (Figure 6A). What should we take from the fact that an oscillator aligned in anti-phase from the with the stimulus (which would presumably show the phase code only stimulus offsets), still shows a near equal correlation with true timing deviations. Furthermore, while the R2 shows that the predictions of the model co-vary with the true values, I'm curious to know how accurately they are predicted overall (in terms of mean squared error for example). Does the model account for deviations from rhythmicity of the right magnitude?

We agree that the differences in R2 depending on the parameters of the oscillation might seem slightly underwhelming. This is likely due to the nature of our fitting. We fit the same parameter (RNN prediction), but use different transformation on this predictor (all of which maintain the same order of the prediction parameter). Therefore, the difference between the model can be viewed as a difference in a transformation applied to the data (the same as a difference between doing for example doing a log transform or an arcsin transformation). In general, OLS is robust to slight variations in these transformations and therefore will still fit at a similar explained variance with only slight variations. Therefore, we believe that these small differences are meaningful but should rather be viewed as a relative comparison than absolute explanation of our different oscillatory parameters. Indeed, this is also why we set all our other parameters of our model to zero (equation 4) and don’t actually simulate stimulus processing, but merely the relative expected shift. We agree with the reviewer that is unlikely that processing at an anti-phase manner would lead to good performance. We demonstrate that in the other figures this is not the case (e.g. processing at anti-phase is not even possible at low stimulus intensities in Figure 5). For this specific comparison we did choose for the OLS and transformation shift and not a brute force fitting as in Figure 8 as (1) we can enter control variables to account for different variance we need to control for in the natural dataset, and (2) on this big dataset will result in a much more efficient code.

To further answer the reviewer’s question, we extracted the mean square error of the model (Author response image 2). See Figure 7 for the original R2 for comparisons. It is evident that the MSE is directly related to the R2 of the model. This is not strange as the lower the error variance, the more the explained variance. Maybe we misunderstood what the reviewer was asking for, but we do not see the direct added benefit of including the MSE value.

4. Lastly, it is unclear to what extent the oscillator is necessary to find this relative time shift. A model comparison between the predictions of the STiMCON and the RNN predictions on their own (à la Figure 3) would help to show how much the addition of the oscillation improves our predictions. Perhaps this is what is meant by the "non-transformed R2" but this is unclear.

We regret that this was unclear. Indeed, we meant with the non-transformed R2 a model with the same factors and same permutations as for the models show figure 6, but not performing the transformation as described in equation 5. This has now been clarified in the manuscript. It now reads: “Results show a modulation of the R2 dependent on the amplitude and phase offset of the oscillation (Figure 7A). This was stronger than a model in which transformation in equation (5) was not applied (R2 was there 0.389).”

5. Figure 7 shows a striking result demonstrating how the model can be used to explain an interesting finding that phase of an oscillation can bias perception towards da or ga. The initial papers consider this result to be explained by delays in onset between visual and auditory stimuli whereas this result explains it in terms of the statistical likelihood each syllable. It is a nice reframing which helps me to better understand the previous result.

We agree that this argument is more parsimonious than our original interpretation. Of course, the two are not mutually exclusive (AV delays could also be bigger for less likely syllables). We have mentioned this now in the result section. It now reads: “Note that previously we have interpreted the /daga/ phase-dependent effect as a mapping of differences between natural audio-visual onset delays of the two syllabic types onto oscillatory phase (Ten Oever et al., 2013, Ten Oever and Sack, 2015). However, the current interpretation is not mutually exclusive with this delay-to-phase mapping as audio-visual delays could be bigger for less frequent syllables.”

6. The authors show that syllable lengths are determined in part by the predictability of the word it is a part of. While the authors have reasonably restricted themselves to a single hierarchical level, the point invites the question as to whether all hierarchical levels are governed by similar processes. Should syllables accelerate from beginning to end of a word? Or in more or less predictable phrases?

We would predict that different level word on similar operations. However, there has not been a great deal of research on syllabic predictions and lengthening or shortening of syllables. In figure 2E we show that indeed words with more syllables are shorten than would be expected by summing up the average syllabic duration expected from mono-syllabic words. However, this does not have to be because later syllables are shortened. Note however that this in not directly what we would predict. We predict that the more predictable the next syllable, the shorter the syllable-to-syllable onsets. So if the next word is very unpredictable, the last syllable would actually be longer. Indeed, often it is found that the last syllable of a word is lengthened (Lindblom, 1968). Other studies show that for individual phonemes word frequency can affect the duration of the syllable (Pluymaekers et al., 2005). But also, higher predictability within the word or based on the next word can influence shortening (Luymakers et al., 2005b). In linguistic studies it has been found that also initial syllables of a word are shortened when words are longer (Lehiste, 1972). While this initially seems against our prediction of higher predictability (within a word) leading to shortening, note that these were only a bi- or tri-syllabic words. For these words we would predict that the first syllable would be shortened when the next syllable is predictable which it the case when a producer can access the full word. In sum, whenever transitional boundary to the next syllable are weaker (across than within words) this can lead to (relative) lengthening (as predicted from Figure 3 and Table 2). This needs to be investigated across syllabic, word, and phrasal levels. However, to fully answer this question we would need to investigate the transitional probabilities of the syllables in the corpus. We have added a section on this point in the discussion. It now reads: “It is likely that predictive mechanisms also operate on these higher and syllabic levels. It is known for example that syllables are shortened when the following syllabic content is known versus producing syllables in isolation (Lehiste ,1972, Pluymaekers et al., 2005). Interactions also occur as syllables within more frequency words are generally shortened (Pluymaekers et al., 2005). Therefore, more hierarchical levels need to be added to the current model (but this is possible following equation (1)).”

7. Figure 5 shows how an oscillator mechanism can force pseudo-rhythmic stimuli into a more rhythmic code. The authors note that this can be done either by slowing responses to early stimuli and quickening responses to later ones, or by dropping (nodes don't reach threshold) stimuli too far outside the range of the oscillation. The first is an interesting mechanism, the second is potentially detrimental to processing (although it could be used as a means for filtering out noise). The authors should make clear how much deviation is required to invoke the dropping out mechanism and how this threshold relates to the naturalistic case. This would give the reader a clearer view of the flexibility of this model.

The dropping mechanism will only every occur when the stimulus intensity is not high enough. For this specific demonstration we kept the stimulus intensity low to demonstrate the change in sensitivity based on feedback. As the current implementation of the model an intensity of 2.1 would be sufficient to always reach activation also at the least excitable point of the oscillation. We have now clarified this in the manuscript. It now reads: “In regular circumstances we would of course always want to process speech, also when it arrives at a less excitable phase. Note however, that the current stimulus intensities were picked to exactly extract the threshold responses. When we increase our intensity range above 2.1 nodes will always get activated even on the lowest excitable phase of the oscillation.”

8. I found Figure 5 very difficult to understand and had to read and read it multiple times to feel like I could get a handle on it. I struggled to get a handle on why supra time was shorter and shorter the later the stimulus was activated. It should reverse at some point as the phase goes back into lower excitability, right? The current wording is very unclear on this point. In addition, the low-high, high-low analysis is unclear because the nature of the stimuli is unclear. I think an added figure panel to show how these stimuli are generated and manipulated would go a long way here.9. The prediction of behavioral data in Figure 7 is striking but the methods could be improved. Currently, the authors bin the output of the model to be 0, 0.5 or 1 which requires some maneuvering to effectively compare it with the sinewave model. They could instead use a continuous measure (either lag of activation between da and ga, or activation difference) as a feature in a logistic regression to predict the human subject behavior.

We have split up the figure in two figure, one relation to Figure 5A+B and the other to C-D-E. In this way we could add clarifying figures on what the model is doing exactly (Figure 5A and Figure 6A). We hope this is now clearer.

Regarding the clarification points.

1) I struggled to get a handle on why supra time was shorter and shorter the later the stimulus was activated. It should reverse at some point as the phase goes back into lower excitability, right?

Indeed, at some point the supra time should reverse, however, we didn’t go so far out as also the feedback would be faded by that time and there would be no difference among the different stimuli.

2) The low-high, high-low analysis is unclear because the nature of the stimuli is unclear.

The nature of the stimuli is now clarified in panel 6A. The stimuli are all the same, but vary in the underlying internal model. For the low-high the stimuli alternate between a highly predicted and not-predicted stimulus and vice versa for high-low.

10. I'm not sure but I think there is a typo in line 383-384. The parameter for feedback should read Cl+1◊ l * Al+1,T. Note the + sign instead of the -. Or I have misunderstood something important.

The lag analysis is difficult as often one of the nodes is not activated at all and we would have to set an arbitrary value to this latency then. But we are able to look at the mean activation for the nodes to get a continuous variable. Therefore, we repeated the analysis but using the relative activation between /da/ and /ga/ nodes over an interval of 500 ms post-stimulus. The results of this analysis are shown in figure 8D. Firstly, the explained variance increases up to 83% compared to the analyses using the first active node as outcome measure. For the rest the pattern is very similar, but for the mean activation, the inhibition function was more important for the strength of the first than using the first active node as outcome. We have decided to keep both types of fitting in our manuscript as we are not sure yet what could be the more relevant neuronal feature for identifying the stimulus. Is it the first time the node is active or the average activation? But as both analyses point to the same direction we are confident that all features of the model are important to fit the data.

References

Assaneo, M. F., et al. (2021). "Speaking rhythmically can shape hearing." Nature human behaviour 5(1): 71-82.

Cumin, D. and C. Unsworth (2007). "Generalising the Kuramoto model for the study of neuronal synchronisation in the brain." Physica D: Nonlinear Phenomena 226(2): 181-196.

Doelling, K. B., et al. (2019). "An oscillator model better predicts cortical entrainment to music." Proceedings of the National Academy of Sciences 116(20): 10113-10121.

Ghitza, O. (2012). "On the role of theta-driven syllabic parsing in decoding speech: intelligibility of speech with a manipulated modulation spectrum." Frontiers in Psychology 3.

Giraud, A. L. and D. Poeppel (2012). "Cortical oscillations and speech processing: emerging computational principles and operations." Nature Neuroscience 15(4): 511-517.

Jensen, O., et al. (2012). "An oscillatory mechanism for prioritizing salient unattended stimuli." Trends in Cognitive Sciences 16(4): 200-206.

Kaufeld, G., et al. (2020). "Linguistic structure and meaning organize neural oscillations into a content-specific hierarchy." Journal of Neuroscience 40(49): 9467-9475.

Lehiste (1972). "The timing of utterances and linguistic boundaries." The Journal of the Acoustical Society of America 51.

Lisman, J. E. and O. Jensen (2013). "The theta-gamma neural code." Neuron 77(6): 1002-1016.

Luo, H. and D. Poeppel (2007). "Phase patterns of neuronal responses reliably discriminate speech in human auditory cortex." Neuron 54(6): 1001-1010.

Meyer, L., et al. (2019). "Synchronous, but not entrained: Exogenous and endogenous cortical rhythms of speech and language processing." Language, Cognition and Neuroscience: 1-11.

Panzeri, S., et al. (2001). "The role of spike timing in the coding of stimulus location in rat somatosensory cortex." Neuron 29(3): 769-777.

Pluymaekers, M., et al. (2005). "Articulatory planning is continuous and sensitive to informational redundancy." Phonetica 62(2-4): 146-159.

Pluymaekers, M., et al. (2005). "Lexical frequency and acoustic reduction in spoken Dutch." The Journal of the Acoustical Society of America 118(4): 2561-2569.

Poeppel, D. and M. F. Assaneo (2020). "Speech rhythms and their neural foundations." Nature Reviews Neuroscience: 1-13.

Rimmele, J. M., et al. (2018). "Proactive sensing of periodic and aperiodic auditory patterns." Trends in Cognitive Sciences 22(10): 870-882.

Schroeder, C. E. and P. Lakatos (2009). "Low-frequency neuronal oscillations as instruments of sensory selection." Trends in Neurosciences 32(1): 9-18.

Ten Oever, S., et al. (2020). "Phase-coded oscillatory ordering promotes the separation of closely matched representations to optimize perceptual discrimination." iScience: 101282.

Ten Oever, S. and A. T. Sack (2015). "Oscillatory phase shapes syllable perception." Proceedings of the National Academy of Sciences 112(52): 15833-15837.

Ten Oever, S., et al. (2013). "Audio-visual onset differences are used to determine syllable identity for ambiguous audio-visual stimulus pairs." Frontiers in Psychology 4.

Winfree, A. T. (2001). The geometry of biological time, Springer Science and Business Media.

Zuidema, W. (2010). "A syllable frequency list for Dutch."

[Editors' note: further revisions were suggested prior to acceptance, as described below.]

Reviewer #2 (Recommendations for the authors):I want to thank the authors for the great effort revising the manuscript. The manuscript has much improved. I only have some final small comments.Detailed commentsl. 273-275: In my opinion: This is because the oscillator is set as a rigid oscillator in the model that is not affected by the word level layer activation; however, as the authors already discuss this topic, this is just a comment.l. 344: "the processing itself" I'd specify: "the processing at the word layer".

We have changed the phrasing accordingly.

l. 557/558: Rimmele et al., (2018) do discuss that besides the motor system, predictions from higher-level linguistic processing might affect auditory cortex neuronal oscillations through phase resetting. Top-down predictions affecting auditory cortex oscillations is one of the main claims of the paper. Thus, this paper seems not a good example for proposals that exclude when-to-what interactions. In my view the claims are rather consistent with the ones proposed here, although Rimmele et al., do not detail the mechanism and differ from the current proposal in that they suggest phase resetting. Could you clarify?

We regret that it seemed that we overlooked that Rimmele et al., (2018) discuss that linguistic predictions can influence auditory cortex. We believe this is a misunderstanding. While Rimmele et al., (2018) indeed discuss how timing in the brain can change due to top-down linguistic predictions, they do not discuss that timing in the speech input itself is also dependent on the content and relates by itself to linguistic predictions. One core question to us is how the brain can extract the statistical information from what-when dependencies in the environment. For example, it is unclear how phase resetting would account for inferring that a less predictable word typically occurs later. This is a core difference between those models and the current model. We realize now that the current phrasing is rather ambiguous talking about when-what in the brain or in the stimulus statistics and the argument was not complete. We rephrase this part of the manuscript. It now reads: “While some of these models discuss that higher-level linguistic processing can modulate the timing of ongoing oscillations [15], they typically do not consider that in the speech signal itself the content or predictability of a word relates to the timing of this word. Phase resetting models typically deal with pseudo-rhythmicity by shifting the phase of ongoing oscillations in response to a word that is off-set to the mean frequency of the input [3, 77]. We believe that this cannot explain how the brain uses what/when dependencies present in the environment to infer the content of the word (e.g. later words are likely a less predictable word).”

l 584 ff.: "This idea diverges from the idea that entrainment should per definition occur on the most excitable phase of the oscillation [3,15]." Maybe rephrase: "This idea diverges from the idea that entrainment should align the most excitable phase of the oscillation with the highest energy in the acoustics [3,15]."

We have changed the phrasing accordingly.

l. 431: "The model consists of four nodes (N1, N2, Nda, and Nga) at which N1 activation predicts a second unspecific stimulus (S2) represented by N2 at a predictability of 1. N2 activation predicts either da or ga at 0.2 and 0.1 probability respectively."This is still hard to understand for me. E.g. What is S2, is this either da or ga, wouldn't their probability have to add up to 1?

N1 and N2 represent nodes that are responsive to two stimuli S1 and S2 (just as Nda and Nga are responsive to the stimulus /da/ and /ga/). S1 and S2 are two unspecific stimuli that are in the simulation to model the entrainment stimuli. (manuscript now reads: “N1 and N2 represent nodes responsive to two stimulus S1 and S2 that function as entrainment stimuli.”)

Indeed, in the brain it would make sense that the prediction adds up to 1. However, we here only model a small proportion of all the possible word nodes in the brain. To clarify this, we added the following: “While in the brain the prediction should add up to 1, we can assume that the probability is spread across a big number of word nodes of the full language model and therefore neglectable.”

Wordingl. 175/176: sth is wrong with the sentence.l. 544: "higher and syllabic"? (sounds like sth is wrong in the wording)l. 546: "within more frequency" (more frequent or higher frequency?)

We have updated the wording of these sentences accordingly.